



Atmospheric
Chemistry
and Physics

# UV spectroscopic determination of the chlorine monoxide (ClO) / chlorine peroxide (ClOOCl) thermal equilibrium constant

**J. Eric Klobas**[1,2] **and David M. Wilmouth**[1,2]

[1]Harvard John A. Paulson School of Engineering and Applied Sciences, Harvard University, Cambridge, MA 02138, USA
[2]Department of Chemistry and Chemical Biology, Harvard University, Cambridge, MA 02138, USA

**Correspondence:** J. Eric Klobas (klobas@huarp.harvard.edu)

**Abstract.** The thermal equilibrium constant between the chlorine monoxide radical (ClO) and its dimer, chlorine peroxide (ClOOCl), was determined as a function of temperature between 228 and 301 K in a discharge flow apparatus using broadband UV absorption spectroscopy. A third-law fit of the equilibrium values determined from the experimental data provides the expression $K_{eq} = 2.16 \times 10^{-27}e^{(8527\pm35\,\mathrm{K}/T)}$ cm$^3$ molecule$^{-1}$ (1$\sigma$ uncertainty). A second-law analysis of the data is in good agreement. From the slope of the van't Hoff plot in the third-law analysis, the enthalpy of formation for ClOOCl is calculated, $\Delta H_f^\circ(298\,\mathrm{K}) = 130.0 \pm 0.6$ kJ mol$^{-1}$. The equilibrium constant results from this study suggest that the uncertainties in $K_{eq}$ recommended in the most recent (year 2015) NASA JPL Data Evaluation can be significantly reduced.

## 1 Introduction

Halogen-mediated catalytic processing of ozone accounts for the overwhelming majority of lower stratospheric ozone-loss processes in polar winter and spring (e.g., WMO, 2014; Wilmouth et al., 2018). Approximately half of this loss (Wohltmann et al., 2017) is resultant from the ClO dimer cycle (Molina and Molina, 1987), which occurs as a result of the highly perturbed physicochemical conditions of the polar vortices:

$$ClO + ClO + M \rightleftharpoons ClOOCl + M, \tag{R1}$$

$$ClOOCl + h\nu \rightarrow ClOO + Cl, \tag{R2}$$

$$ClOO + M \rightarrow Cl + O_2 + M, \tag{R3}$$

$$2(Cl + O_3 \rightarrow ClO + O_2), \tag{R4}$$

$$net: 2O_3 \rightarrow 3O_2. \tag{R5}$$

Within this cycle, the equilibrium governing the partitioning of ClO and ClOOCl in Reaction (R1) is defined as follows.

$$K_{eq} = \frac{[ClOOCl]}{[ClO]^2} \tag{1}$$

This thermal equilibrium is a key parameter that determines the nighttime partitioning of active chlorine in the winter–spring polar vortex. The value of $K_{eq}$ can also tune the efficiency of chlorine-mediated ozone destruction, particularly the radial extent of ozone loss within the warmer Arctic polar vortex. For example, Canty et al. (2016) quantified how small variations in $K_{eq}$ can modulate significant changes in the temperature at which photolysis of ClOOCl and thermal decomposition of ClOOCl occur at equal rates.

Although the partitioning between ClO and ClOOCl is highly important, relatively few laboratory measurements of $K_{eq}$ have been made, and there is significant disagreement between reported values. Accordingly, the uncertainty in $K_{eq}$ was large (e.g., $\sim 75\%$ at 200 K) as of the 2011 JPL compendium recommendation (Sander et al., 2011). The most recent 2015 JPL-recommended value of $K_{eq}$ was revised on the basis of a 2015 study by Hume et al. (2015), but the recommended uncertainties are still substantial, exceeding 50% at 200 K (Burkholder et al., 2015).

The preponderance of laboratory data from previous determinations of $K_{eq}$ was obtained at temperatures significantly warmer than the polar stratosphere ($T > 250$ K). Error in the extrapolation of these warm temperature data has often been cited to explain the lack of correspondence between values of $K_{eq}$ determined in the laboratory and those calcu-

lated from stratospheric observations (Avallone and Toohey, 2001; Stimpfle et al., 2004; von Hobe et al., 2005; Santee et al., 2010). The more recent results of Hume et al. (2015) are unique in that they were obtained at temperatures colder than other laboratory studies (206 K < $T$ < 250 K), but their experimental method was compromised by secondary bimolecular reactions at warmer temperatures. In the present study, our spectroscopic data bridge the warmer temperatures where most laboratory determinations of $K_{eq}$ have been made to the colder-temperature work of Hume et al. (2015), covering a broader temperature range than any previous study. The thermal equilibrium constant between ClO and ClOOCl was measured as a function of temperature (228 K < $T$ < 301 K) by UV spectroscopy and is evaluated here in relation to prior determinations, observations, and recommendations from compendia CE3.

## 2 Experiments

All experiments were conducted in a discharge flow apparatus, as shown in Fig. 1. Independently programmable thermal zones allow for the optimization of target chemistry as a function of flow velocity, temperature, and pressure. ClO is synthesized via the reaction of Cl atoms with $O_3$ (Reaction R4). Cl is produced from a 1 % $Cl_2$/He gas mixture, diluted further with UHP He and directed through a 45 W, 2.45 GHz microwave discharge. $O_3$ is produced via electric discharge of a 10 % $O_2$/Ar source mixture and subsequently introduced 2.5 cm after the microwave cavity. Once formed, ClO readily dimerizes to form ClOOCl (Reaction R1), particularly at higher concentrations and colder temperatures.

To facilitate dimerization of ClO, the gas mixture is cooled in a 20 cm long jacketed quartz cell immediately subsequent to the microwave discharge. This reaction cell (Fig. 1) has an inner diameter of 1 cm and can be maintained at a temperature between 198 and 305 K via circulating chilled methanol (NESLAB Endocal ULT-80). The operation of this cell at cold temperatures additionally suppresses undesired chemistry, preventing the synthesis of side products such as OClO per Reaction (R6), and subsequently $Cl_2O_3$ per Reaction (R7).

$$ClO + ClO \rightarrow OClO + Cl \tag{R6}$$

$$ClO + OClO + M \rightarrow Cl_2O_3 + M \tag{R7}$$

Following the reaction cell, the gas stream then passes through the cold trap zone, which is maintained at temperatures between 100 K and room temperature depending on the experiment. Cooling is accomplished by flowing $N_2$ gas through a copper coil immersed in liquid $N_2$ and then through an 18 cm long insulated aluminum jacket surrounding the 1 cm inner diameter quartz flow tube. Type K thermocouples (alumel–chromel) are affixed at three positions on the outside of the flow tube, opposite the cryogenic gas ports. These thermocouples are further insulated to ensure the recorded volt-

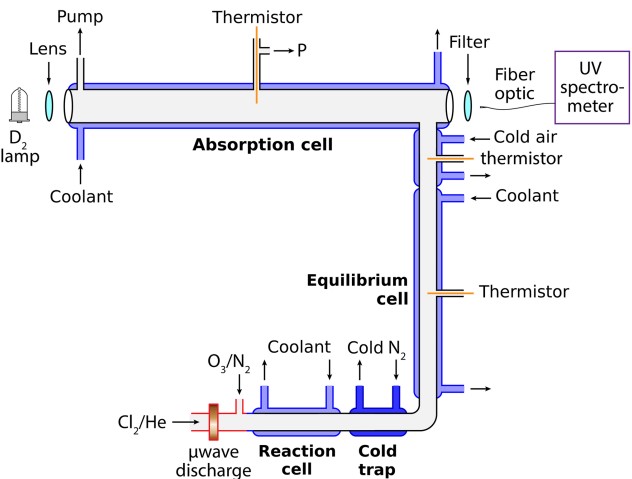

**Figure 1.** Schematic of the discharge-flow absorbance experiment. Dilute chlorine gas in helium flows through a microwave discharge to form Cl radicals. Dilute ozone in nitrogen is then injected to produce ClO radicals. Self-reaction of ClO occurs in the cold reaction cell to form ClOOCl. When utilized, the cold trap provides for halogen oxide purification. ClO/ClOOCl equilibrium is established in the equilibrium cell, which is held at the same temperature as the absorption cell. The gas mixture is then characterized via UV spectroscopy in the absorption cell using software developed in-house.

ages correspond to the temperature of the quartz tube and not the temperature of the cryogenic gas. The coupling between the cold trap and the equilibrium cell is actually linear but is presented as a right angle in Fig. 1 for graphical purposes.

The next section of the flow system in Fig. 1, labeled equilibrium cell, is a jacketed 50 cm quartz tube of 1 cm inner diameter. This section is where the gases reach equilibrium prior to measurement in the absorption cell. The equilibrium cell and the absorption cell share a coupled circulating chilled methanol bath (NESLAB ULT-80), ensuring that the two cells are maintained at the same temperature. The equilibrium cell is isolated from the environment with two 10 mm blankets of aerogel insulation (Cryogel Z). Additionally, a flow of cryogenic $N_2$ passes through an insulated aluminum jacket surrounding the union between the equilibrium cell and the absorption cell. This $N_2$ is chilled by passing through a copper coil immersed in the reservoir of the circulating chiller servicing the reaction cell, and the flow is modulated to provide constant temperature as the gas mixture transits from the equilibrium cell to the detection axis. A 100 Ω thermistor is inserted into the gas stream at this location to verify the temperature.

Finally, the gas mixture enters the absorption cell, a 91.44 cm jacketed quartz tube with an inner diameter of 2.54 cm. This detection axis is oriented at a right angle to the equilibrium cell and is terminated with two quartz windows. A 100 Ω thermistor is positioned at the halfway point. Cryogenic circulating methanol provides for temperature control

between 228 and 301 K. Two 10 mm blankets of aerogel insulation (Cryogel Z) provide thermal isolation from the environment. An exterior dry $N_2$ purge is employed to prevent window condensation.

The discharge reactor is operated at pressures between 100 and 333 mbar. Pressure is monitored with Baratron capacitance manometers. Carrier gas flow rates, $\sim 1.0$–$1.8$ L min$^{-1}$ depending on the experimental conditions, are metered via MKS mass flow controllers. $Cl_2$ flow rates are controlled via a needle valve, while $O_3$ addition is modulated using micrometer flow control valves. Total system pressure and velocity are tuned using an integral bonnet needle valve. Residence times within the absorption cell range between $\sim 1$ and 11 s, depending on gas flow rates, system temperature, and pressure. A quarter-turn plug valve provides a bypass of the integral bonnet needle valve such that rapid pump-down of the reactor and reignition of the plasma can be performed without disturbing pressure calibration during the course of the experiments.

Data were acquired using a fiber-coupled Ocean Optics USB4000 UV-Vis spectrometer ($\sim 0.3$ nm resolution) illuminated by a Hamamatsu L2D2 deuterium lamp. The need to correlate Baratron, thermocouple, and thermistor sensor readings with each UV spectrum required the in-house development of custom software. Drivers and libraries to operate the spectrometer and simultaneously interrogate analog sensors were written in Python 2.7 and, in combination with the Python-seabreeze CE4 library, provided scriptable, automated control of nearly all aspects of the data acquisition system.

The deuterium lamp was allowed to warm up for at least 1 h prior to data collection activity to reduce small variations in lamp output on experimental timescales. Dark spectra were acquired prior to any experiments on a daily basis. Background spectra were obtained with the microwave plasma extinguished and all gas flows of species that absorb in the region of 200–295 nm TS2 (e.g., $O_3$, $Cl_2$) off. For consistency, sample spectra were obtained exactly 100 s after the background spectra against which they were referenced. Each saved spectrum consists of the coaddition of 597 individual scans, the number of scans that could be obtained in exactly 3 min of acquisition time.

To aid in the selection of experimental conditions, a simulation of the discharge-flow reactor was constructed. A numerical integrator for chemical kinetics (written in Python 2.7 with NumPy and SciPy) for 18 chemical species and 45 relevant chemical reactions was informed by JPL Data Evaluation 15-10 kinetic rate constants (Burkholder et al., 2015) and coupled into a physical model of gas flows as a function of reactor geometry, temperature, and pressure. Temperature and pressure ranges were scanned to determine optimal conditions to ensure ClO–ClOOCl equilibrium within the real-world experiment. Because parameterized simulations carry inherent uncertainty, experimental conditions were selected at several pressures along the equi-

librium asymptote ($K_{eq}$ vs. $P$), and real-world experiments were performed at pressures above and below the identified value in order to confirm asymptotic equilibrium behavior. The kinetic model was only used to inform conditions for the experimental setup, but no results from the model were used in the determination of the reported equilibrium constants.

## 3 Results and discussion

More than 136 000 background and sample spectra were obtained between the temperatures of 228 and 301 K at pressures ranging between 100 and 333 mbar. Typical initial concentrations spanned $2 \times 10^{13}$– TS3 $4 \times 10^{14}$ molecules cm$^{-3}$ for $O_3$ and $1 \times 10^{14}$–$4 \times 10^{15}$ molecules cm$^{-3}$ for $Cl_2$. Active chlorine (ClO$_x$) concentrations were typically $1 \times 10^{13}$–$1 \times 10^{14}$ molecules cm$^{-3}$ with the microwave discharge on. These values were tuned according to the initial conditions prescribed by the model simulations, as described above. For example, as the target temperature of the experiment decreased, the system was operated at incrementally higher pressures to allow more time for equilibrium to be achieved. Higher temperature samples reached equilibrium more readily, so gas velocity was increased to limit the impact of enhanced rates of secondary chemistry on observed $K_{eq}$ values.

Multicomponent spectral curve fitting software packages were programmed in Python 2.7/LmFit (Newville et al., 2016) for the deconvolution of the UV absorption spectra of $O_3$, $Cl_2$, ClO, ClOOCl, OClO, and $Cl_2O_3$. Reference cross sections were utilized as follows: for $O_3$ and OClO, pure sample spectra were acquired and scaled to match the 2015 JPL-recommended cross sections of Molina and Molina (1986) and Kromminga et al. (2003), respectively. ClOOCl and $Cl_2O_3$ cross sections were obtained directly from the 2015 JPL Data Evaluation (Burkholder et al., 2015). Temperature-dependent cross sections of $Cl_2$ were obtained from Marić et al. (1993) and validated to match observed $Cl_2$ spectra along the experimental temperature range. Synthetic temperature-dependent cross sections from Marić and Burrows (1999) were used for ClO due to the broad temperature range over which the data are available.

The cross sections from Marić and Burrows (1999) were found to provide an excellent fit of experimentally obtained ClO at all relevant temperatures in this study and were validated against available laboratory-determined ClO cross sections from the literature. Our experimental spectra at 263 K fit using both the synthetic ClO cross sections of Marić and Burrows (1999) and the reported experimental cross sections of Trolier et al. (1990) at 263 K result in concentrations of ClO that differ by only 3.0 %. Similarly, experimental samples at 300 K from this study fit to the room-temperature laboratory ClO cross sections of Simon et al. (1990) and Sander and Friedl (1988) have excellent correspondence with the synthetic cross sections of Marić and Burrows (1999): 1.1 % deviation in ClO concentration in comparison with Si-

mon et al. (1990) and 2.6 % deviation in comparison with Sander and Friedl (1988). The resolution of our experimental spectra was degraded to match the lower-resolution data of Trolier et al. (1990) for this comparison, while the cross sections of Sander and Friedl (1988) and Simon et al. (1990), which were published at higher resolution than provided by our spectrometer, were degraded to our spectral resolution.

A total of 82 experimental observations of the equilibrium gas mixture were acquired across a broad range of conditions (e.g., changing initial gas concentrations, pressure, temperature, carrier gas flow rates, or microwave discharge power). Multiple observations acquired under the same experimental conditions were reduced to a single measurement by coaddition of spectra. Samples at colder temperatures were subjected to more repeated evaluations to improve accuracy under the low ClO conditions.

Concentrations of ClO and ClOOCl were obtained via spectral deconvolution as follows. Quantification of ClO was determined from high-pass filtration of each absorbance spectrum to remove the contributions of absorbers that are spectrally smooth. The highly structured vibrational bands of the ClO $A^2\Pi \leftarrow X^2\Pi$ transition were similarly extracted from the ClO reference spectra at the appropriate temperature, and least-squares minimization was conducted in the wavelength region of 260–300 nm. ClOOCl was then quantified by multicomponent linear regression, constraining ClO to the previously determined concentration, over the wavelength range of 230–260 nm TS4. The custom software used to deconvolve the measured spectra used a differential evolution minimizer via a stochastic process; to test the reproducibility of the deconvolution and to generate fit statistics, each spectral fit was run 100 times. The component gas concentrations for each sample were then determined as the average of the entire deconvolution ensemble for that sample. Though not employed in the calculation of $K_{eq}$, OClO was also quantified by spectral deconvolution in the wavelength region of 310–350 nm, where instrumental sensitivity to OClO is maximized. Only in three experimental runs, at temperatures of 294 K and above, was the concentration of OClO greater than the concentration of the two species of interest – and even then, not by a large margin. $Cl_2O_3$ was below the detection limit for all spectra collected in this study. The resulting concentrations of ClO, ClOOCl, and OClO are enumerated in Table 1. Also shown are the uncertainties ($1\sigma$) for each component determined from the spectral fitting procedure, which do not exceed 8 % for ClO and 18 % for ClOOCl and in most cases are $< 2$ % for ClO and $< 5$ % for ClOOCl. Using the vibrational structure to define the spectral fits for ClO significantly reduces the uncertainty.

Figure 2 demonstrates the fitting process for an experiment conducted at 268.7 K. The raw spectrum and the wavelength regions used to fit each component are presented in panel (a). The colored traces overlapping the raw spectrum show the fit results, with each trace being the sum of the deconvolved components. Additionally, the residual from each fit is shown, offset from the baseline for clarity and provided at 3-times magnification for additional diagnosis of fit quality. Except for a slight deviation in the OClO fit at the longest wavelengths as a result of source instability in this low-light region, the residuals for all three fitting techniques are observed to be flat across the wavelength ranges and minimally structured. This holds true for all spectral fits reported in this work. High-pass filtration of ClO is depicted in panel (b), wherein differential absorbance data are plotted with the high-pass-filtered ClO component from the fit overlaid. The quantification of ClOOCl is presented in panel (c), in which the absorbances of ClOOCl and $O_3$ are plotted on a log scale for improved visibility, along with the experimental spectrum for reference. Finally, panel (d) shows the OClO and $Cl_2$ fitted spectral components between 310 and 350 nm, again plotted on a log scale with the experimental spectrum shown for reference. We note that OClO is a very small component of the absorbance spectrum and is near the instrumental detection limit, especially in the wavelength regions employed for the quantification of ClO and ClOOCl.

Once the ClO and ClOOCl concentrations are determined from the spectral fit, the value of $K_{eq}$ at the relevant temperature is calculated per Eq. (1). $K_{eq}$ values for each sample are shown in Table 1 and plotted in Fig. 3a as a function of inverse temperature.

The temperature dependence of $K_{eq}$ can be related as an Arrhenius expression, per Eq. (2), with free parameters $A$ and $B$.

$$K_{eq} = Ae^{(B/T)} \tag{2}$$

In a third-law fit, the prefactor $A$ is fixed to a prescribed value. For this work, we employ the JPL Data Evaluation recommended $A$ parameter value of $2.16 \times 10^{-27}$ cm$^3$ molecule$^{-1}$, which is the most recent literature evaluation of this constant (Burkholder et al., 2015). A third-law fit of our $K_{eq}$ data yields a $B$ parameter value of 8527 K. This result was obtained by an ordinary least-squares fit of the 82 measurements of $K_{eq}$ (Table 1). The fit is shown in Fig. 3a as the black trace. Error from the fitting process was quantified via bootstrapping with 2000 resamplings of the binned $K_{eq}$ results, which establishes a fit error interval of $\pm 5.2$ K. This method only accounts for the fit error and does not take into account other potential sources of experimental error, as discussed below.

The accuracy of the spectral deconvolution demonstrated a temperature dependence due to ClO or ClOOCl concentrations approaching their experimental limit of quantification (ClO limiting measurements at colder temperatures and ClOOCl at warmer temperatures). There is also the potential for secondary chemistry to impact the $K_{eq}$ results, particularly at warmer temperatures (Reaction R6). To assess the significance of these temperature-dependent factors, least-squares third-law fits were performed on sub-sampled data populations. Specifically, an analysis of $K_{eq}$ results obtained

**Table 1.** Experimental conditions[a], concentrations from spectral deconvolution[b], and $K_{eq}$ values[c].

| $T$ (K) | $\frac{[\text{ClO}]}{10^{11}}$ | $\frac{[\text{ClOOCl}]}{10^{11}}$ | $\frac{[\text{OClO}]}{10^{11}}$ | $K_{eq}$ | $T$ (K) | $\frac{[\text{ClO}]}{10^{11}}$ | $\frac{[\text{ClOOCl}]}{10^{11}}$ | $\frac{[\text{OClO}]}{10^{11}}$ | $K_{eq}$ |
|---|---|---|---|---|---|---|---|---|---|
| 300.7 | 244 (0.3) | 30.9 (13) | 27.3 (0.01) | $5.19 \times 10^{-15}$ | 253.7 | 41.2 (2) | 158 (5) | – | $9.31 \times 10^{-13}$ |
| 297.8 | 128 (0.4) | 10.3 (9) | 15.2 (1) | $6.29 \times 10^{-15}$ | 250.7 | 43.3 (1) | 244 (2) | – | $1.30 \times 10^{-12}$ |
| 297.6 | 127 (0.4) | 9.77 (8) | 16.3 (0.5) | $6.06 \times 10^{-15}$ | 250.6 | 36.2 (2) | 154 (1) | – | $1.18 \times 10^{-12}$ |
| 294.0 | 126 (0.4) | 11.7 (7) | 16.3 (0.5) | $7.37 \times 10^{-15}$ | 250.6 | 41.2 (2) | 251 (3) | – | $1.48 \times 10^{-12}$ |
| 294.0 | 126 (0.3) | 15.4 (6) | 15.3 (1) | $9.70 \times 10^{-15}$ | 250.5 | 39.7 (2) | 243 (2) | – | $1.54 \times 10^{-12}$ |
| 292.7 | 140 (0.2) | 23.6 (6) | 14.4 (1) | $1.20 \times 10^{-14}$ | 250.4 | 25.0 (4) | 90.7 (8) | – | $1.45 \times 10^{-12}$ |
| 291.1 | 132 (0.3) | 17.9 (10) | 15.6 (0.6) | $1.03 \times 10^{-14}$ | 250.4 | 23.7 (4) | 74.7 (1) | – | $1.33 \times 10^{-12}$ |
| 291.1 | 133 (0.3) | 25.3 (8) | 15.6 (0.8) | $1.43 \times 10^{-14}$ | 250.4 | 26.8 (2) | 72.5 (1) | – | $1.01 \times 10^{-12}$ |
| 291.0 | 133 (0.3) | 22.7 (3) | 16.7 (0.8) | $1.28 \times 10^{-14}$ | 250.1 | 37.4 (2) | 198 (5) | – | $1.42 \times 10^{-12}$ |
| 285.2 | 124 (0.3) | 33.3 (6) | 13.6 (1) | $2.17 \times 10^{-14}$ | 250.0 | 35.0 (1) | 171 (2) | – | $1.40 \times 10^{-12}$ |
| 285.0 | 120 (0.3) | 35.4 (5) | 13.7 (2) | $2.46 \times 10^{-14}$ | 249.9 | 36.0 (2) | 160 (6) | – | $1.23 \times 10^{-12}$ |
| 277.8 | 88.4 (0.6) | 37.6 (7) | 7.56 (3) | $4.81 \times 10^{-14}$ | 249.9 | 35.1 (1) | 165 (5) | – | $1.34 \times 10^{-12}$ |
| 277.8 | 86.2 (0.7) | 23.5 (3) | 8.41 (1) | $3.16 \times 10^{-14}$ | 245.4 | 31.7 (2) | 258 (2) | – | $2.57 \times 10^{-12}$ |
| 277.8 | 90.1 (0.5) | 30.3 (5) | 8.78 (2) | $3.73 \times 10^{-14}$ | 245.2 | 29.7 (2) | 246 (2) | – | $2.79 \times 10^{-12}$ |
| 277.8 | 92.6(0.6) | 47.0 (2) | 7.66 (3) | $5.48 \times 10^{-14}$ | 244.9 | 29.9 (2) | 282 (3) | – | $3.15 \times 10^{-12}$ |
| 275.3 | 109 (0.4) | 80.8 (2) | 8.50 (1) | $6.80 \times 10^{-14}$ | 244.9 | 31.5 (2) | 285 (3) | – | $2.87 \times 10^{-12}$ |
| 275.2 | 108 (0.3) | 72.4 (10) | 8.41 (2) | $6.21 \times 10^{-14}$ | 244.9 | 28.7 (2) | 266 (3) | – | $3.23 \times 10^{-12}$ |
| 275.1 | 110 (0.3) | 71.1 (8) | 7.59 (2) | $5.88 \times 10^{-14}$ | 240.6 | 26.8 (3) | 367 (1) | – | $5.11 \times 10^{-12}$ |
| 275.1 | 109 (0.4) | 72.3 (5) | 7.65 (3) | $6.09 \times 10^{-14}$ | 240.6 | 27.1 (2) | 368 (3) | – | $5.01 \times 10^{-12}$ |
| 268.8 | 77.2 (0.6) | 72.2 (8) | 4.15 (4) | $1.21 \times 10^{-13}$ | 238.7 | 17.5 (4) | 164 (3) | – | $5.36 \times 10^{-12}$ |
| 268.8 | 74.7 (0.6) | 70.0 (9) | 3.87 (6) | $1.25 \times 10^{-13}$ | 238.6 | 7.88 (4) | 51.8 (11) | – | $8.34 \times 10^{-12}$ |
| 268.7 | 73.8 (0.8) | 81.1 (9) | 4.54 (3) | $1.49 \times 10^{-13}$ | 238.6 | 18.0 (5) | 185 (3) | – | $5.71 \times 10^{-12}$ |
| 268.7 | 73.4 (0.6) | 66.8 (9) | 4.48 (4) | $1.24 \times 10^{-13}$ | 237.1 | 6.49 (7) | 37.6 (18) | – | $8.93 \times 10^{-12}$ |
| 265.5 | 78.5 (0.5) | 78.4 (7) | 2.82 (5) | $1.27 \times 10^{-13}$ | 236.9 | 7.38 (8) | 44.0 (18) | – | $8.08 \times 10^{-12}$ |
| 265.3 | 76.7 (0.5) | 129 (6) | 3.11 (5) | $2.19 \times 10^{-13}$ | 236.5 | 15.1 (4) | 210 (2) | – | $9.21 \times 10^{-11}$ |
| 265.2 | 109 (0.8) | 202 (2) | – | $1.70 \times 10^{-13}$ | 236.5 | 14.5 (6) | 223 (4) | – | $1.06 \times 10^{-11}$ |
| 265.2 | 111 (0.8) | 236 (5) | – | $1.92 \times 10^{-13}$ | 236.2 | 15.0 (4) | 238 (4) | – | $1.06 \times 10^{-11}$ |
| 265.2 | 78.1 (0.6) | 117 (7) | 2.26 (6) | $1.92 \times 10^{-13}$ | 235.3 | 10.3 (4) | 112 (4) | – | $1.06 \times 10^{-11}$ |
| 265.2 | 77.8 (0.6) | 127 (6) | 2.15 (8) | $2.10 \times 10^{-13}$ | 235.1 | 21.2 (4) | 342 (3) | – | $7.61 \times 10^{-12}$ |
| 265.1 | 106 (0.8) | 189 (5) | – | $1.68 \times 10^{-13}$ | 234.1 | 13.6 (3) | 259 (1) | – | $1.40 \times 10^{-11}$ |
| 265.0 | 104 (0.8) | 216 (2) | 1.17 (25) | $2.00 \times 10^{-13}$ | 234.1 | 15.3 (4) | 286 (3) | – | $1.22 \times 10^{-11}$ |
| 260.4 | 56.0 (0.9) | 126 (4) | 1.19 (14) | $4.02 \times 10^{-13}$ | 234.1 | 14.2 (4) | 310 (3) | – | $1.54 \times 10^{-11}$ |
| 260.2 | 55.6 (1) | 110 (5) | 1.43 (8) | $3.56 \times 10^{-13}$ | 234.1 | 15.6 (3) | 283 (2) | – | $1.16 \times 10^{-11}$ |
| 260.1 | 54.1 (0.9) | 115 (6) | 1.09 (9) | $3.93 \times 10^{-13}$ | 233.2 | 7.38 (6) | 77.1 (6) | – | $1.42 \times 10^{-11}$ |
| 260.0 | 53.4 (0.9) | 120 (5) | 1.02 (8) | $4.21 \times 10^{-13}$ | 231.6 | 12.4 (4) | 366 (2) | – | $2.38 \times 10^{-11}$ |
| 254.7 | 45.7 (2) | 211 (4) | – | $1.01 \times 10^{-12}$ | 231.4 | 10.9 (3) | 400 (2) | – | $3.37 \times 10^{-11}$ |
| 254.6 | 47.7 (2) | 207 (4) | – | $9.10 \times 10^{-13}$ | 230.8 | 12.3 (6) | 365 (2) | – | $2.41 \times 10^{-11}$ |
| 254.5 | 45.6 (1) | 170 (4) | – | $8.18 \times 10^{-13}$ | 230.5 | 10.4 (6) | 246 (3) | – | $2.27 \times 10^{-11}$ |
| 254.5 | 44.5 (1) | 178 (4) | – | $8.99 \times 10^{-13}$ | 230.4 | 7.18 (7) | 135 (5) | – | $2.62 \times 10^{-11}$ |
| 254.0 | 44.6 (2) | 166 (4) | – | $8.35 \times 10^{-13}$ | 230.4 | 7.92 (7) | 173 (2) | – | $2.76 \times 10^{-11}$ |
| 253.7 | 43.1 (2) | 165 (3) | – | $8.88 \times 10^{-13}$ | 228.1 | 5.36 (6) | 120 (7) | – | $4.18 \times 10^{-11}$ |

[a] Experimental pressures: $T > 292$ K, $P = 100$ mbar; $280$ K $< T < 292$ K, $P = 133$ mbar; $270$ K $< T < 280$ K, $P = 166$ mbar; $261$ K $< T < 270$ K, $P = 200$ mbar; $252$ K $< T < 261$ K, $P = 233$ mbar; $242$ K $< T < 252$ K, $P = 266$ mbar; $T < 242$ K, $P = 333$ mbar. [b] Concentration units: molecules cm$^{-3}$. Dashes indicate quantity below limit of detection. Parenthetical values provide percentage uncertainty in concentration from the spectral deconvolution. [c] $K_{eq}$ cm$^3$ molecule$^{-1}$.

at 250–301, 228–291, and 250–291 K resulted in $B$ parameters of 8531, 8525, and 8530 K, respectively, which are in excellent agreement with the parameter of 8527 K obtained from a fit of the entire temperature range. The extrapolated value of $K_{eq}$ at 200 K obtained from these sub-sampled data sets varies by $< 3\%$, while at 180 K the spread of the maximal deviation in $K_{eq}$ between subsets is $< 4\%$. An estimate of the third-law error from temperature-dependent precision is $\pm 3$ K about the $B$ parameter.

The reproducibility of the $B$ parameter value regardless of the temperature range employed in the spectral fits provides strong evidence that secondary chemistry does not signifi-

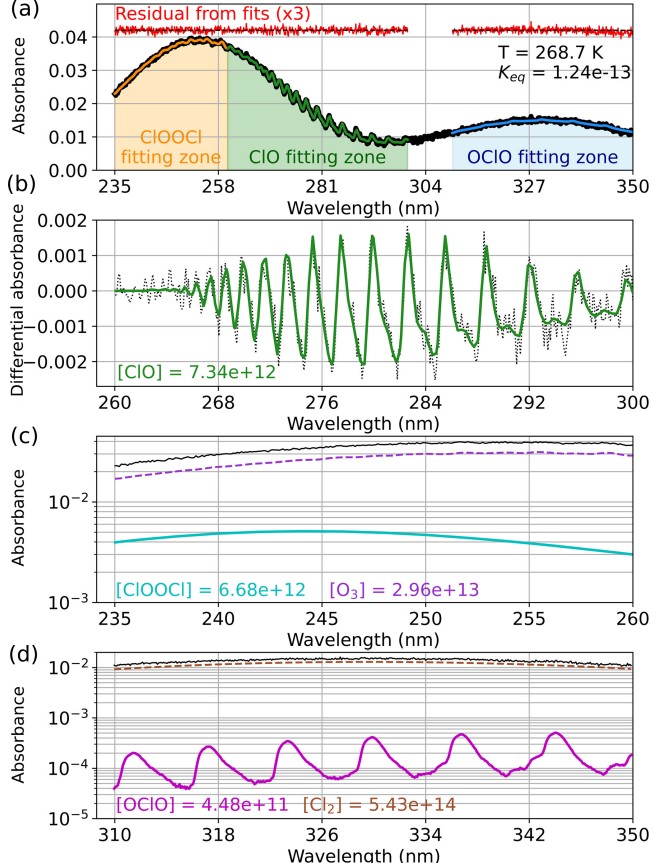

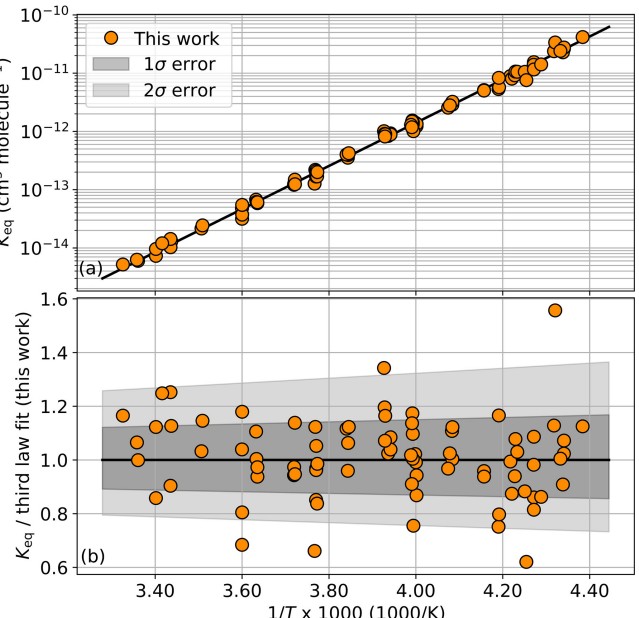

**Figure 3. (a)** Experimental $K_{eq}$ values as a function of $1000/T$. The black trace is a third-law fit of the orange circles, $K_{eq} = 2.16 \times 10^{-27} e^{(8527\,K/T)}$ cm$^3$ molecule$^{-1}$. Individual values are enumerated in Table 1. **(b)** Ratio of experimental $K_{eq}$ values to the third-law fit. Dark gray shading encompasses the total estimated $1\sigma$ error from this study and light shading encompasses the $2\sigma$ error.

**Figure 2.** Deconvolution of experimental absorbance spectra to component contributions for an experiment conducted at 268.7 K. **(a)** Experimental spectrum (black) and the wavelength ranges used for quantification of each spectral component (orange: 235–260 nm, green: 260–300 nm, blue: 310–350 nm). The resulting spectral fits for each region are overlaid on the experimental spectrum in colored traces; the residuals from the fits are plotted in red, magnified by a factor of 3, and offset for clarity. The obtained $K_{eq}$ (cm$^3$ molecule$^{-1}$) is indicated in the top right corner. **(b)** Quantification of ClO by the high-pass-filtration method. The differential absorbance spectrum appears as black dots, and the fitted ClO component is depicted in solid green. **(c)** Quantification of ClOOCl (cyan) and O$_3$ (purple) with the experimental spectrum shown for reference (black). **(d)** Quantification of OClO (magenta) and Cl$_2$ (brown) with the experimental spectrum shown for reference (black). Note that only the primary absorbing species are shown in panels **(c)** and **(d)**, but all relevant absorbers are included in the fits. Fitted concentrations (molecules cm$^{-3}$) are given for various component gases.

cantly impact the $K_{eq}$ values reported here. To further confirm this result, least-squares third-law fits were performed separately on all of the experimental runs in Table 1 in which OClO concentrations were large enough to be quantified and on all of the runs in Table 1 in which OClO concentrations were below our limit of detection. The resulting $B$ values

with and without OClO are essentially identical, 8526 and 8527 K, respectively.

Variation in the reference cross section of one component in a multicomponent fit may impact the quality of fit for the other spectral components. It is the trend in the literature to not explicitly include uncertainties from the reference cross sections during assignation of error for $K_{eq}$; however, the choice of synthetic cross sections for ClO is considered further here. As discussed, the synthetic temperature-dependent ClO cross sections prepared by Marić and Burrows (1999) were employed for the determination of [ClO] in this study, and the ClO concentrations from fits using these synthetic cross sections differed from ClO concentrations determined using experimentally derived reference standards at most by 3.0 %. To capture the uncertainty of this error, all ClO concentrations used to derive $K_{eq}$ were scaled by $\pm 3.0$ % and then fit by a least-squares third-law analysis, producing an estimated error in $B$ due to ClO cross section selection of $\pm 15$ K.

Ultimately, estimating our uncertainty on the $B$ parameter from all known sources of potential error yielded comparable, but smaller, values than simply assigning the error interval such that it fully encompassed > 68 % of the individual $K_{eq}$ results at $1\sigma$. Accordingly, we assign error intervals ($1\sigma$) to our $B$ parameter of $\pm 35$ K. Systematic errors arising from experimental design and postprocessing techniques are estimated to contribute errors that sum to a total smaller than this

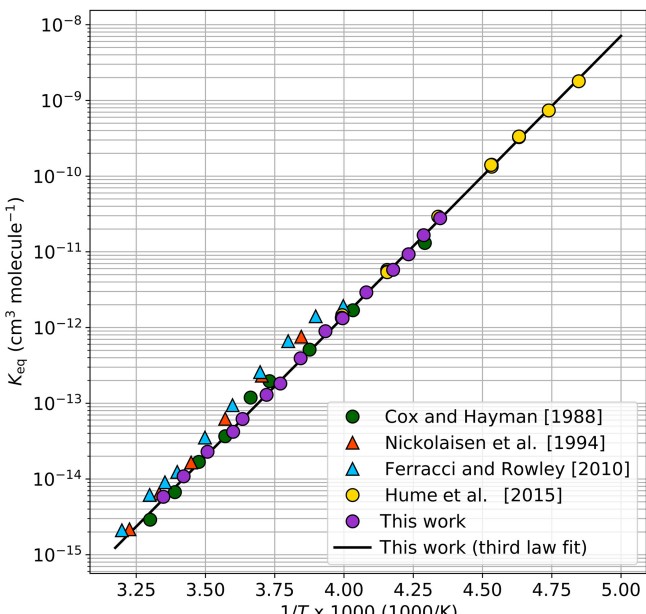

**Figure 4.** $K_{eq}$ values as a function of $1000/T$. The black trace is the third-law fit as determined in Fig. 3, $K_{eq} = 2.16 \times 10^{-27} e^{(8527\,\mathrm{K}/T)}\,\mathrm{cm}^3\,\mathrm{molecule}^{-1}$. For clarity, data from this work are plotted as 3 K averages (purple circles) of the full data set shown in Fig. 3. The colored markers are $K_{eq}$ values from prior laboratory studies as reported in the literature.

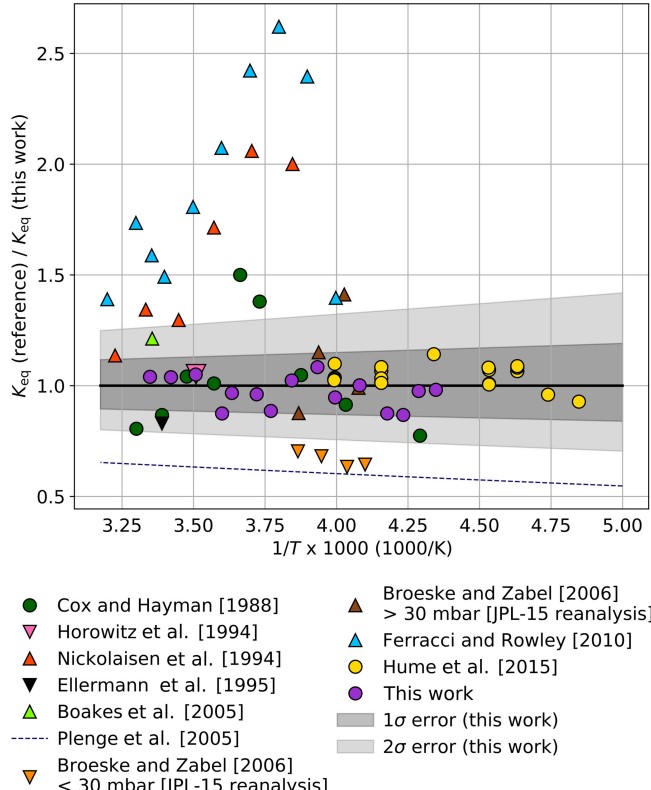

**Figure 5.** Ratio of $K_{eq}$ values from prior laboratory studies to $K_{eq}$ determined in this work ($K_{eq} = 2.16 \times 10^{-27} e^{(8527\,\mathrm{K}/T)}\,\mathrm{cm}^3\,\mathrm{molecule}^{-1}$) as a function of $1000/T$. Circles indicate studies in which $K_{eq}$ was measured with UV spectroscopy of equilibrium mixtures, while triangles indicate works in which $K_{eq}$ was determined from individual reaction kinetic rates. For clarity, data from this work are plotted as 3 K averages (purple circles) of the full data set shown in Fig. 3. Dark gray shading encompasses the total estimated $1\sigma$ error from this study and light shading encompasses the $2\sigma$ error.

boundary. The ratio of the individual $K_{eq}$ measurements to the resulting $K_{eq}$ expression from the third-law fit is depicted in Fig. 3b along with the estimate of error. From this figure, it can be seen that 71 % of the individual $K_{eq}$ measurements reside within the $1\sigma$ uncertainty interval. The resulting $K_{eq}$ expression from the third-law analysis is provided in Eq. (3).

$$K_{eq} = 2.16 \times 10^{-27} e^{(8527 \pm 35\,\mathrm{K}/T)}\,\mathrm{cm}^3\,\mathrm{molecule}^{-1} \qquad (3)$$

Figure 4 provides a comparison of our results to equilibrium constants determined from other laboratory studies. In this and subsequent figures, our data are shown averaged in 3 K intervals, the minimal spacing required to ensure that each data point is averaged with at least one other data point. The presentation of averaged data is for improved figure clarity only; all data analyses were performed with the $K_{eq}$ expression derived from the full data set (Eq. 3). Notably in Fig. 4, the $K_{eq}$ values reported in this work are typically smaller than prior evaluations of $K_{eq}$ at warmer temperatures but match well with the colder temperature observations of Hume et al. (2015) for those data points with overlap (228–250 K). Moreover, our third-law fit matches well with the results of Hume et al. (2015) when extrapolated to 200 K.

The ratio of $K_{eq}$ values from prior laboratory studies relative to $K_{eq}$ calculated from Eq. (3) is shown in Fig. 5. The $1\sigma$ ($B\pm35$ K) and $2\sigma$ ($B\pm70$ K) error bounds from this work are plotted as shaded gray tones. The $K_{eq}$ values from previous laboratory studies were derived either using UV absorption

spectroscopy of equilibrium mixtures of ClO and ClOOCl or by determination of the individual forward and/or reverse kinetic rates of dimerization, Reaction (R1). Experimental data from these previous studies are shown as circles and triangles, respectively. As evident in Fig. 5, there is much greater variation in the determinations of $K_{eq}$ using kinetics methods.

The experiments of Cox and Hayman (1988) and Hume et al. (2015) were performed using UV analysis. The results of Hume et al. (2015) lie entirely within our $1\sigma$ error interval. Though the individual results of Cox and Hayman exhibit significant scatter and some measurements exceed the $2\sigma$ error reported here, an ordinary least-squares third-law fit of their results using the JPL-recommended $A$ parameter (Burkholder et al., 2015) remains within our $1\sigma$ error boundaries ($B = 8537$ K).

Nickolaisen et al. (1994) and Ferracci and Rowley (2010) used flash photolysis/UV absorption spectroscopy to determine the kinetic rates of the individual reactions in order to determine $K_{eq}$. Though these two studies agree with each other in trend and magnitude, they both exhibit significant departures from our results, possibly due to secondary reactions given the high concentrations of ClO and Cl$_2$O employed in those studies.

Bröske and Zabel (2006) investigated the kinetics of the ClOOCl dissociation reaction and estimated $K_{eq}$ values using JPL 2002 kinetics (Sander et al., 2003) for the forward Reaction (R1). A reanalysis of their results using JPL 2015 kinetics (Burkholder et al., 2015) is plotted as binned averages in Fig. 5 (orange triangles, $P < 30$ mbar; brown triangles, $P > 30$ mbar). A reanalysis of the high-pressure results of Bröske and Zabel (2006) also provides a third-law fit ($K_{eq} = 2.16 \times 10^{-27} e^{(8498\,\mathrm{K}/T)}$ cm$^3$ molecule$^{-1}$) that resides within our 1$\sigma$ error limits. The discrepancy between the experiments of Bröske and Zabel (2006) conducted at higher pressures and lower pressures is discussed in depth in their work.

Horowitz et al. (1994) examined the loss rate of ClO while monitoring the kinetics and branching ratio of the ClO + ClO reaction and provide a single-point estimate of $K_{eq}$ at 285 K that is within 1$\sigma$ of the value determined at that temperature in this work. The $K_{eq}$ values of Boakes et al. (2005) using flash photolysis/UV absorption spectroscopy and Ellermann et al. (1995) using pulsed radiolysis/UV absorption spectroscopy lie within our 2$\sigma$ uncertainty; however, the values of Plenge et al. (2005) via mass spectrometric determination of the ClO–OCl bond strength lie outside the 2$\sigma$ error limits from this work.

Figure 6 provides a comparison between observational determinations of $K_{eq}$ in the atmosphere and an extrapolation of $K_{eq}$ from this work to 190 K. The determination of $K_{eq}$ by Avallone and Toohey (2001), an analysis of AASE I and AASE II data in which mixing ratios of ClOOCl were inferred from total Cl$_y$ mass conservation rather than directly measured, agrees within error with the results of this work. Similarly, a determination of $K_{eq}$ by Santee et al. (2010), informed by ClO mixing ratios retrieved via Aura MLS satellite data and ClOOCl mixing ratios calculated from stratospheric modeling, lies in substantial agreement with the $K_{eq}$ expression derived here. All observed ratios of [ClOOCl]/[ClO]$^2$ in the Arctic stratosphere for the nighttime ER-2 flight of 3 February 2000 during the SOLVE/THESEO campaign (Stimpfle et al., 2004) also agree within the combined 2$\sigma$ measurement uncertainties (SOLVE/THESEO measurement uncertainties not plotted for clarity). The $K_{eq}$ expression from von Hobe et al. (2005), which was based on observations in Arctic winter 2003, deviates substantially from $K_{eq}$ determined in this work; however, von Hobe et al. (2007) postulate that those previous observations of ClO and ClOOCl may not have been in equi-

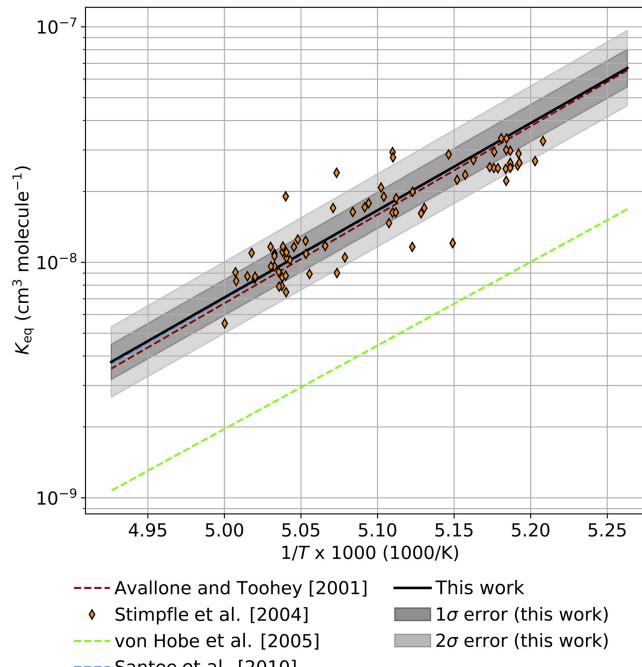

**Figure 6.** Comparison of extrapolated $K_{eq}$ values from the third-law fit in this work ($K_{eq} = 2.16 \times 10^{-27} e^{(8527\,\mathrm{K}/T)}$ cm$^3$ molecule$^{-1}$) to atmospheric observations. $K_{eq}$ (solid black) and error boundaries (gray shaded regions) determined in this study are extrapolated to the temperature range of 190–203 K. Expressions for $K_{eq}$ derived from previous atmospheric measurements are presented as dashed lines. Observations of [ClOOCl]/[ClO]$^2$ from the nighttime ER-2 flight on 3 February 2000 in the SOLVE/THESEO mission out of Kiruna, Sweden, are indicated as orange diamonds.

librium and that the ClOOCl measurements may have been biased low.

A thermodynamic representation of parameters $A$ and $B$ (Eq. 2) can be obtained from a manipulation of the van't Hoff equation. The prefactor $A$ encodes the standard entropy of reaction change per Eq. (4), in which the superscript indicates a standard state of 1 bar; $R'$ is the gas constant (83.145 cm$^3$ bar mol$^{-1}$ K$^{-1}$); $N_A$ is Avogadro's constant ($6.0221 \times 10^{23}$ molecules mol$^{-1}$); $R$ is the gas constant in energy units (8.3145 J mol$^{-1}$ K$^{-1}$); $e$ is Euler's number; and $T$ is system temperature CE5 .

$$\Delta S^\circ (298\,\mathrm{K}) = R \ln \left( \frac{N_A A}{e R' T} \right) \qquad (4)$$

The exponential argument $B$ relates the change in standard enthalpy of reaction as shown in Eq. (5), with $R$ and $T$ as defined above.

$$\Delta H^\circ (298\,\mathrm{K}) = -R (T + B) \qquad (5)$$

Evaluating Eq. (5) with our derived value of $B = 8527$ K results in $\Delta H^\circ (298\,\mathrm{K})$ of $-73.4 \pm 0.6$ kJ mol$^{-1}$ for Reac-

tion (R1). The uncertainty estimate on this value was obtained by combining the previously determined uncertainty in our $B$ parameter ($\pm 35$ K) with estimated uncertainties in the reference cross sections for ClOOCl ($\pm 17\%$ variation near the peak cross section at 248 nm, as reported by Lien et al., 2009; Papanastasiou et al., 2009; and Wilmouth et al., 2009) and for ClO ($\pm 3\%$ variation in the fitted concentrations of ClO between Marić and Burrows, 1999; Sander and Friedl, 1988; Simon et al., 1990; and Trolier et al., 1990) to produce a possible range in $B$ of 8450 to 8603 K, as determined from scaled, third-law least-squares fits.

Combining our $\Delta H^\circ(298\,\mathrm{K})$ value for Reaction (R1) with the JPL-recommended $\Delta H_f^\circ(298\,\mathrm{K})$ for ClO of $101.681 \pm 0.040\,\mathrm{kJ\,mol^{-1}}$ (Burkholder et al., 2015) yields a $\Delta H_f^\circ(298\,\mathrm{K})$ for ClOOCl of $130.0 \pm 0.6\,\mathrm{kJ\,mol^{-1}}$. This result is in excellent agreement with the JPL-recommended value of $130.1 \pm 1\,\mathrm{kJ\,mol^{-1}}$.

A least-squares second-law fit of $K_{eq}$, in which both $A$ and $B$ are free parameters, yields a determination of $K_{eq}$ as shown in Eq. (6).

$$K_{eq} = (2.70 \pm 0.6)$$
$$\times 10^{-27} e^{(8470 \pm 60\,\mathrm{K}/T)}\,\mathrm{cm^3\,molecule^{-1}} \quad (6)$$

The uncertainties from the second-law fit are larger but the results agree well with the third-law fit, e.g., agreement to within 3 % at 298 K and to within 6 % at 200 K. An application of Eq. (4) to the second-law prefactor of $2.70 \times 10^{-27}\,\mathrm{cm^3\,molecule^{-1}}$ produces $\Delta S^\circ(298\,\mathrm{K}) = -145.8^{+1.7}_{-2.1}\,\mathrm{J\,mol^{-1}\,K^{-1}}$ for Reaction (R1), which agrees with the JPL-recommended value of $-147.0\,\mathrm{J\,mol^{-1}\,K^{-1}}$ (calculated from the $S$ values for ClO and ClOOCl in Table 6-2 of Burkholder et al., 2015). The value of $\Delta H^\circ(298\,\mathrm{K})$ of $-72.9 \pm 1.0\,\mathrm{kJ\,mol^{-1}}$ for Reaction (R1) from the second-law analysis is in agreement with our results from the third-law analysis.

Notably, the equilibrium constant results obtained in this work agree in trend and magnitude with the recently reported $K_{eq}$ values of Hume et al. (2015). This excellent correspondence is illustrated in Fig. 7, in which a least-squares third-law fit (with each study weighted equally) is presented for a combined data set containing the results of this work and the work of Hume et al. (2015) in ratio to the current JPL recommendation of $K_{eq} = 2.16 \times 10^{-27} e^{(8537\,\mathrm{K}/T)}\,\mathrm{cm^3\,molecule^{-1}}$ (Burkholder et al., 2015). The combined works span a temperature range of 206–301 K, and the resulting $K_{eq}$ is $2.16 \times 10^{-27} e^{(8532\,\mathrm{K}/T)}\,\mathrm{cm^3\,molecule^{-1}}$. This expression deviates from the JPL-recommended $K_{eq}$ value at 200 K by 2.5 %. For illustrative purposes, the uncertainty bounds calculated from this work and the bounds recommended by the current JPL evaluation (Burkholder et al., 2015) are also plotted. Note that all of the plotted data lie within our $1\sigma$ uncertainty; this is because our data are averaged in 3 K intervals here, but our uncertainty was determined from the variance in our full data set (Table 1). The JPL uncertainty, which was not de-

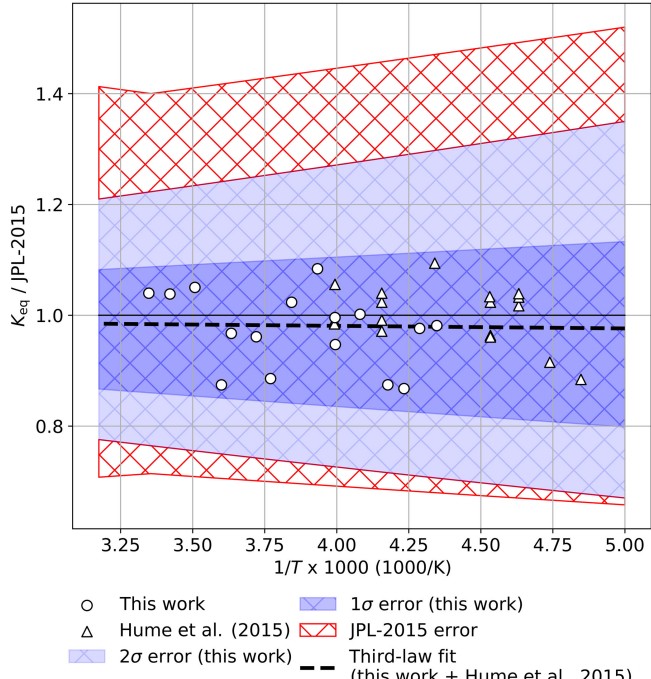

**Figure 7.** Ratio of a weighted third-law fit of $K_{eq}$ (black dashed line) determined from a combination of this work (circles) and Hume et al. (2015) (triangles) to the JPL compendium recommended value (Burkholder et al., 2015). For clarity, data points from this work are plotted as 3 K averages of the full data set shown in Fig. 3. Error intervals as reported in this work (darker blue = $1\sigma$, lighter blue = $2\sigma$) and as recommended by JPL-2015 (red cross hatch).

rived from a statistical analysis, but was scaled to encompass the warm temperature results of Cox and Hayman (1988) and Nickolaisen et al. (1994) and the low temperature work of Hume et al. (2015), greatly exceeds the scatter of the individual $K_{eq}$ values from the combined data set of this work and Hume et al. (2015). Our results suggest that the uncertainties in the current JPL recommendation for $K_{eq}$ can be reduced.

## 4 Conclusions

The thermal equilibrium governing the association of ClO and dissociation of ClOOCl was investigated in a custom-built discharge-flow reactor by UV spectroscopy between the temperatures of 228 and 301 K. The selected temperature range allowed us to bridge the warmer temperature regime where nearly all previous laboratory studies of $K_{eq}$ have been performed and the recent colder temperature work of Hume et al. (2015). A third-law fit of our $K_{eq}$ results deviates from some prior laboratory studies but demonstrates excellent agreement with the work of Hume et al. (2015) and with the currently recommended parameters in the JPL com-

pendium (Burkholder et al., 2015). The agreement between our third-law and second-law analyses lends further confidence to the results reported herein. Our calculated enthalpy of formation for ClOOCl from the slope of the van't Hoff plot is in excellent agreement with the recommended value (Burkholder et al., 2015).

The current JPL-recommended error bounds for the ClO–ClOOCl equilibrium constant are large (Burkholder et al., 2015), exceeding 50 % at 200 K. The excellent correspondence between the $K_{eq}$ results from this work and Hume et al. (2015) lends confidence to the established parameterization of the JPL Data Evaluation (Burkholder et al., 2015), suggesting that prescribed error intervals for this reaction can be reduced.

*Data availability.* Data used in this publication are presented in Table 1 and can also be accessed by request to the corresponding author.

*Author contributions.* JEK and DMW designed, constructed, and operated the experiment and analyzed and reported the data described in this work.

*Competing interests.* The authors declare that they have no conflict of interest.

*Acknowledgements.* We gratefully acknowledge funding from the National Aeronautics and Space Administration (NASA) through grants NNX15AF60G, NNX15AD87G, and 80NSSC18K1063. We thank Marco Rivero and Norton Allen for engineering support.

*Review statement.* This paper was edited by James B. Burkholder and reviewed by Marc von Hobe, John Barker, Darin Toohey, and one anonymous referee.

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

**Remarks from the language copy-editor**

**Remarks from the typesetter**