# Peer review of "UV spectroscopic determination of the chlorine monoxide (ClO) / chlorine peroxide (ClOOCl) thermal equilibrium constant"

_Atmospheric Chemistry and Physics, 2018_

## Referee Comment (RC1) · J. R. Barker (Referee) · 7 Nov 2018

This paper describes experimental measurements of the equilibrium constant (Keq) for the gas phase reaction ClO + ClO = ClOOCl at temperatures from 228 to 301 K. The measurements were performed by using wavelength-resolved UV spectroscopy to simultaneously determine the equilibrium ClO and ClOOCl concentrations. Great care was taken to identify the chemical species by their absorption spectra and to establish accurate absorption cross sections. The results for Keq are in excellent agreement with the recent experimental measurements of Hume et al. (2015). The two stud-
ies overlap in temperature to some extent and the present work extends the range to higher temperatures. (Both sets of data are in reasonable agreement with previous studies, which were not as precise and were also subject to interference from secondary reactions.) Together, the two studies constitute a set of internally consistent high-precision measurements of this important equilibrium constant over much of the atmospheric temperature range of interest. Thermodynamic parameters (entropy and enthalpy changes) were obtained from both a second-law and a third-law analysis of the data. The two approaches are in good agreement with each other and the thermodynamic parameters are in good agreement with literature values.

This is a highly important subject. The paper is very well written and the methods are clearly explained. The atmospheric context and relation of the new results to literature data are described exceptionally well.

A quibble with the third-law analysis carried out in this paper is that it assumes that $K_{eq}$ is a pure exponential function, $K_{eq} = A \exp(B/T)$, while theoretical calculations predict that $K_{eq}$ deviates slightly from a pure exponential. The approach taken in this paper differs from the third-law analysis carried out by the NASA/JPL Panel for Data Evaluation. The Panel evaluates every individual data point to determine the reaction enthalpy change at 0 K, $\Delta H_r(0)$, extracted from the measured $K_{eq}(T)$ at each temperature and then determines the average. $\Delta H_r(298)$ is then computed from the calculated reaction entropies and average $\Delta H_r(0)$. From $\Delta H_r(298)$, one can then obtain the standard enthalpy of formation, $\Delta H_f(298)$, for ClOOCl. Although more laborious, the approach followed by the panel does not assume a pure exponential function and is thought to be more accurate. From the Panel's approach, the combined data set consisting of the new data from the present paper and the older data from Hume et al. (2015) gives $\Delta H_r(298) = 130.1$ kJ/mol, which is 0.2 kJ/mol higher than the result reported in the manuscript, based on the pure exponential function.

In the abstract, it would be useful to state that the experimental errors are $\pm 1\ \sigma$.

In addition to the binned values for Keq(T), which are reported in Table 1, all of the individual Keq measurements should be reported in Supporting Information.

In Figures 4 and 6, the error bounds are functions of temperature. It would be useful to report the functional forms and parameters for the error bounds in Supporting Information.

---

## Referee Comment (RC2) · 29 Nov 2018

The paper describes a well-designed laboratory study to measure the ClO/ClOOCl thermal equilibrium constant over a broad temperature range and includes a comprehensive analysis and comparison to earlier work. Based on the advanced methodology, broader temperature range and thorough thermodynamic analysis compared to earlier studies, I support publication in ACP.

Before submitting a final version for publication, I strongly recommend that the authors

to add some more information on their experiments and to drop the last sentence of the abstract because I don't think that the significant reduction of uncertainties is warranted. My reasoning for these suggestions is given below.

1. Add more experimental information

I have little doubt that the authors have carefully checked the conditions of each experiment to ensure that thermal equilibrium between ClO and ClOOCl is actually established once the gas mixture passes into the absorption cell. Nevertheless, for the sake of transparency, it would be good to provide some actual numbers for flow rates and residence times in the different parts of the apparatus.

Ideally, a complete set of experimental conditions (initial concentrations, pressure, $T_{ReactionCell}, T_{ColdTrap}, T_{EquilibriumCell}$) and selected results (maybe even some raw spectra) could be provided as an electronic supplement that goes beyond the summary given in Table 1.

2. Uncertainty assessment

I am not convinced that all potential sources of uncertainty are represented in the significantly reduced overall uncertainty presented in Figure 6. In particular, I'm thinking about the 17 % uncertainty in the ClOOCl reference cross sections that you mention on page 8, line 14. When I translate the uncertainty in the B parameter that accounts for this (given on page 8, line 17), I arrive at approximately +/- 40 % uncertainty in $K_{eq}$, which is considerably larger than the uncertainty range given in Figure 6. There, it looks as if the blue uncertainty ranges only represent the statistical uncertainties in your experiment (i.e. the scatter between the individual white circles in that figure represented in the +/- 25 uncertainty in B given in Equation 3), and the statement about the trend to not explicitly include uncertainties from reference cross sections on page 6, line 21, seems to support that interpretation.

In the work by Hume et al. (2015), there is a clear statement that the systematic

uncertainties in cross sections to convert $K_{abs}$ to $K_{eq}$ are much more important than their experimental standard deviations, and their error propagation calculations arrive at an upper uncertainty limit from this is 36 % on the experimental $K_{eq}$, which appears to be a very realistic number.

Note that the 17 % uncertainty in the ClOOCl cross section alone translates directly into a 17 % uncertainty in the ClOOCl concentration fitted to a particular spectrum, and in turn into a systematic 17 % uncertainty in $K_{eq}$ via your Equation 1. The two studies by Papanastasiou et al. (2009, currently the JPL recommended value and used in both, your study and Hume et al.) and Lien et al. (2009; 17 % higher) used different methods to measure absolute cross sections: Papanastasiou et al. infer the absolute value from the reaction stoichiometry and the experimental absorbance at isosbestic wavelength, while Lien et al. use a known quantity of light to attenuate a molecular beam of ClOOCl. To date, no convincing evidence has been presented that proves either one of the results right or wrong. Therefore, the 17 % is not a statistical one or two sigma uncertainty, but a highly systematic one. As long as this is not resolved, it is therefore impossible to reduce the uncertainty in $K_{eq}$ below this value with any method relying on the ClOOCl cross sections. And because it is systematic, it really has to be added to the blue ranges representing the statistical uncertainties in your Figure 6, because if the peak ClOOCl absorption cross section was 17 % higher than the value you actually used in your calculations, all your points and the ones from Hume et al. would simultaneously down by 17 %. Of course, if the uncertainties in the ClOOCl cross sections were reduced in future studies, this would immediately also reduce the uncertainty in $K_{eq}$, which I think would be a fair and valuable statement to make.
* * *

---

## Referee Comment (RC3) · Toohey (Referee) · 3 Dec 2018

General Comments

Early observations under perturbed ozone hole conditions found that ClO was non-zero in polar darkness, a result that was explained by the equilibrium with a weakly-bound adduct of ClO. A body of subsequent work showed that this species was chlorine peroxide, ClOOCl. Photolysis of ClOOCl to liberate chlorine atoms, while preserving the O-O bond, is the rate-determining step in the catalytic scheme proposed explain sudden and rapid loss of ozone over Antarctica. It is now known that similar chemistry occurs in the Arctic and in exhaust plumes of rockets fueled by ammonium perchlorate. The research group at Harvard has proposed that this chemistry may be important over the continental United States in summertime. The competition between ozone-destroying photolysis of ClOOCl and harmless thermal decomposition of ClOOCl is a strong function of temperature, largely determined by the magnitude of the equilibrium constant for exchange between ClO and ClOOCl. Roughly speaking, when the temperature in the stratosphere is greater than about 230 K, thermal decomposition of ClOOCl begins to dominate over photolysis, and the effectiveness of catalytic destruction of ozone by chlorine is rapidly diminished. Therefore, in order to account for ozone destruction by chlorine quantitatively, especially above 220 K, one needs a value for Keq that is very accurate.

There are additional reasons why careful laboratory measurements of Keq are important, in particular the key role in linking observations of ClO to the budget of reactive chlorine, the difficulty of measuring ClOOCl in the atmosphere with sufficient accuracy to constrain ozone loss calculations, and demonstration of the fundamental chemical mechanism for ozone destruction under chlorine-activated conditions.

This paper describes a careful and thorough measurement of the ClO/ClOOCl equilibrium over the temperature range ∼230 K to 300 K using a newly designed, moderate-pressure discharge flow reactor coupled to an absorption cell for direct measurements of ClO and ClOOCl using well-characterized UV spectroscopy. This was no easy task, and the authors are to be commended for their high level of preparation, care, and execution of the experiment that is evidenced by the high precision of the results (i.e., near-linear van't Hoff relationship) and closeness to previous results. A key takeaway message is that combined with another recent study (by Hume et al., 2015 - see reference in paper), recommendations of Keq values for use in atmospheric modeling can be improved; in particular, uncertainties in those recommendations at temperatures relevant for stratospheric ozone loss can be reduced. I agree, for the most part, with

this conclusion, but because this is the main conclusion of this paper the authors have a little more work to do to make a solid case. My specific comments follow.

Specific Comments

I find the overall presentation of experimental methods, schematic of the apparatus, and figures supporting the conclusions to be clear, adequate, and (for the most part) necessary. However, there were several points where I was craving additional detail in order to convince myself that the uncertainties have been evaluated adequately.

The authors need to present more details of their data and error analysis, clearly addressing precision (statistical errors), accuracy, and possible systematic errors. It is possible that they have adequately addressed all these in their final assessment, but because the main point of the paper is that this study can reduce the current errors assigned to Keq by the JPL assessment panel, they need to spend more time and focus on this aspect of the study. I highly recommend adding a few elements to the paper:

1. Please show an enlargement of residual spectra (e.g., as in Figure 2) with examples calculated at the 1-sigma ranges of the uncertainties. This is especially important for the results at lowest temperatures where ClO absorbances are smallest relative to total absorbance due to the [ClO]-squared nature of Keq.

2. I think it would be very useful to plot the results of the Keq calculations versus inverse temperature of individual replicates prior to temperature averaging. This will help illustrate the reproducibility of the runs (i.e., precision) and give a better sense for the density of replicates at specific temperatures. Note, also, that it is more appropriate to average log(Keq) values from individual replicates than to average Keq values (as described on Page 5, line 25) to avoid a systematic bias of a few percent due to the exponential nature of Keq.

3. Because of the strong dependence of Keq on temperature, a more detailed description of temperature variations and accuracy is essential. Please show (or describe)

[Figure]

how temperature varies axially and radially within the measurement cell during a given replicate. Also, it would be useful to know how much temperature varies with time over the course of a particular replicate. I am a little concerned that a single-point measurement of temperature in the center of a measurement cell may not be adequate for a quantitative assessment of uncertainties (e.g., note that a 1.0 degree variation in temperature translates into a 15% variance in Keq). Presumably the uniformity of temperature has been carefully measured and documented at various temperatures. If so, presentation of such evidence will greatly strengthen the case that this new measurement can be used to reduce uncertainties in the JPL assessment.

4. I appreciate the rationale for co-varying pressure and temperature; however, given the possibility of systematic biases due to pressure (e.g. secondary reactions), it would be useful to know if any detailed measurements with varying pressure were carried out for a fixed temperature. On Page 6, starting on Line 10, the authors state "The precision of repeated measurements conducted at the same temperature but varied flow rates and pressures did not statistically deviate from the precision from temperature-dependence alone." Over what range (or percentage) were flow rates and pressures varied for a fixed temperature? Or does this refer to unintended variations that may have occurred over the course of a particular replicate?

5. The authors need to show a more detailed error analysis that traces the various sources of error (e.g., from spectral fitting, temperature, and errors in rate parameters for interfering secondary reactions, if relevant). They should also include an assessment of potential systematic errors (such as those described above). They could expand Table 1 to include these errors. I am not sure that the standard deviation values listed in Table 1 are uniformly illustrative - for example, there is no way that a 0.3% standard deviation from two independent measurements at 285.1 K is representative of the true precision when the standard deviation is 10% for the 8 replicates at 253.3 K.

Minor comments

6. Abstract/conclusions. The authors should report the value of Keq over the temperature range 230 to 299 K, reflecting the range over which they have calculated their experimental averages. Alternatively, if they want to claim significance for a measurement at 288 K then they should report a value that is measured over the range 285.5 to 290.5 K (assuming a similar 5 degree average).

7. Page 2, line 28. Please elaborate on "...optimization of target chemistry." What, specifically, was optimized?

8. Page 2, lines 30-31. Discuss whether or not you expect discharge of oxygen to produce O2(singlet delta), and if so, how you might expect reactions of this specie to impact your results.

9. Page 3, line 30. Please list your carrier gas flow rates and residence times in each of the cells.

10. Page 7, lines 31-33. It might be helpful to include a representative 1 sigma uncertainty bar on the results from the February 3, 2000, SOLVE/THESEO ER-2 flight in Figure 5. Please note whether "measurement uncertainties" for those data points refer to uncertainties (or variability) in measured concentrations of ClO and ClOOCl, uncertainties (or variability) of measured temperature, or both. This could also be illustrated with the use of vertical (for concentrations) and horizontal (for temperature) error bars.

11. Page 9, line 8. You might elaborate on how, specifically, the uncertainties in JPL recommended Keq can be reduced. Should results of previous experiments be discounted by the JPL panel? Or should results of various experiments over the years be averaged and weighted according to errors reported at the time?
* * *

---

## Referee Comment (RC4) · Anonymous Referee #4 · 5 Dec 2018

This manuscript reports spectroscopic measurements of the concentrations of [ClO] and [ClOOCl] under conditions that should be very close to equilibrium, allowing the equilibrium constant to be determined as a function of temperature. There are several items, mentioned below, that should be addressed before this manuscript is accepted for publication.

The microwave discharge that generates atomic chlorine will also be a strong source of VUV radiation due to the strong Cl resonance lines. Since the ozone is introduced only 2.5 cm downstream, one would expect significant photolysis of the ozone, generating

[Figure]

O(1D), O(3P), O2(a1delta), etc. Also the discharge that generates ozone will also form copious amounts of the singlet states of O2. Does the chemical modeling of this system show that none of these reactive species are causing trouble downstream? Typical gas flow rates or the velocities of the flows at the different pressures should be given.

Page 4, line 5: What is the resolution (FWHM) for this slit width?

Page 4, line 10: What are "dark spectra" and how are they used in subsequent spectra?

The two runs shown in Fig. 2 can be used to calculate two equilibrium constants: for 230K, K = 1.84E(-11) which is 34% lower than Equation 3 predicts; for 300K, K = 6.52E(-15) which is 34% higher than Eq.3. Are these typical deviations?

The treatment of the experimental data by "binning" is not the best use of the experimental data. If I understand this manuscript correctly, all values of K measured within a 5 K range of temperatures are averaged (average of K, or ln(K)?) and then listed in Table 1 along with the averaged temperature (average T or 1/T?). For example, if measurements were done at 250 and 255 K, one should get, according to Equation 3, K(250) = 1.438E(-12) and K(255) = 7.364E(-13). The average of ln(K) of these two K's gives -27.602 or K(ave) = 1.029E(-12), which is close to the value predicted by Eq. 3 for T = 252.5 K. But then using the deviations of these two values of K from the average K to get an estimate of the standard deviation of the measurements is not valid. There is no information about the random measurement errors in these two numbers. They are different because they were measured at two different temperatures. Supposed one has only two measurements in one bin that were are different temperatures but, due to random errors, gave almost the same value for K. The calculated "standard deviation" would now be very small and the weighting factor very large. That is not right. The numbers in the last column in Table 1 should be eliminated.

It would be better to do the least squares fitting using all 114 experimental measurements with equal weighting. Then the deviations of the experimental K values from the least squares fit would give information about the precision of the measurements. A

plot of these 114 deviations vs. 1/T would be useful. As mentioned, this will probably show increasing deviations at both the upper and lower limits of temperature. Certainly the experimental values of the 114 measurements should be preserved, either in a table in the manuscript or as supplemental material.

With attention to the above comments, this manuscript should be accepted for publication.

---

## Author Comment (AC1) · 31 Jan 2019

We would like to thank the referee for his detailed and constructive comments and have revised the manuscript accordingly. The reviewer's comments are presented below in **bold text** and our responses to the reviewer appear in plain text.

**1. A quibble with the third-law analysis carried out in this paper is that it assumes that $K_{eq}$ is a pure exponential function, $K_{eq} = A exp^{(B/T)}$, while theoretical calculations predict that $K_{eq}$ deviates slightly from a pure exponential. The**

[Figure]

**approach taken in this paper differs from the third-law analysis carried out by the NASA/JPL Panel for Data Evaluation. The Panel evaluates every individual data point to determine the reaction enthalpy change at 0 K, $\triangle$H$_r$(0), extracted from the measured $K_{eq}(T)$ at each temperature and then determines the average. $\triangle$H$_r$(298) is then computed from the calculated reaction entropies and average $\triangle$H$_r$(0). From $\triangle$Hr(298), one can then obtain the standard enthalpy of formation, $\triangle$H$_f$(298), for ClOOCl. Although more laborious, the approach followed by the panel does not assume a pure exponential function and is thought to be more accurate. From the Panel's approach, the combined data set consisting of the new data from the present paper and the older data from Hume et al. (2015) gives $\triangle$H$_r$(298) = 130.1 kJ/mol, which is 0.2 kJ/mol higher than the result reported in the manuscript, based on the pure exponential function.**

We appreciate the approach described by the reviewer and the accuracy of this method for use in the NASA/JPL Data Panel Evaluation. However, to maintain consistency between our results and the method employed in prior papers (e.g., Hume et al. 2015), we have chosen to use the enthalpy values at 298 K for the calculations. Note that we now determine $B$ from our unbinned data, and the calculated $\triangle$H$_f$(298 K) for ClOOCl is 130.0 kJ/mol, now in even better agreement with the JPL-15 recommendation of 130.1 kJ/mol.

To facilitate alternate derivations of reaction enthalpies, all independent measurements are now reported in Table 1.

**2. In the abstract, it would be useful to state that the experimental errors are $\pm 1\sigma$.**

The abstract now reads: A third law fit of the equilibrium values determined from the experimental data provides the expression:

$K_{eq} = 2.16 \times 10\text{-}27 e^{(8528 \pm 25 K/T)} cm^3 molecule^{-1}(1\sigma \text{ uncertainty})$.

**3. In addition to the binned values for $K_{eq}(T)$, which are reported in Table 1, all of the individual $K_{eq}$ measurements should be reported in Supporting Information.**

All independent measurements are now reported in Table 1.

**4. In Figures 4 and 6, the error bounds are functions of temperature. It would be useful to report the functional forms and parameters for the error bounds in Supporting Information.**

The error bounds of Figure 4 in the original manuscript (now Figure 5) are of the form: $Ae^{(B\pm\epsilon/T)}$ from equation 2, where $\varepsilon$ is the appropriate uncertainty interval. We now explicitly mention the $\epsilon$ value on page 7 line 9 of the revised manuscript.

---

## Author Comment (AC2) · 31 Jan 2019

We would like to thank the referee for his detailed and constructive comments and have revised the manuscript accordingly. The reviewer's comments are presented below in **bold** text and our responses to the reviewer appear in plain text.

**1. Add more experimental information**
**I have little doubt that the authors have carefully checked the conditions of each experiment to ensure that thermal equilibrium between ClO and ClOOCl is ac-**

[Figure]

tually established once the gas mixture passes into the absorption cell. Never-theless, for the sake of transparency, it would be good to provide some actual numbers for flow rates and residence times in the different parts of the appa-ratus. Ideally, a complete set of experimental conditions (initial concentrations, pressure, $T_{ReactionCell}$, $T_{ColdTrap}$, $T_{EquilibriumCell}$) and selected results (maybe even some raw spectra) could be provided as an electronic supplement that goes be-yond the summary given in Table 1.

Table 1 has now been revised to include all of the experimental runs and not simply the binned data, and details on carrier gas flows and residence time have been added in section 2. We note that a critical element of the experimental design and operation was ensuring that ClO and ClOOCl were in equilibrium in the absorption cell when the measurements were made. As discussed in the manuscript, the kinetic model defined the experimental conditions (e.g., temperatures, pressures, concentrations) in which equilibrium would be achieved in the laboratory setup, and we operated experimen-tally across a range of conditions about the optimal starting conditions predicted by the model. In particular, we note that the same $K_{eq}$ values were obtained when increas-ing or decreasing residence times in the equilibrium and absorption cells, providing confidence that ClO and ClOOCl were indeed in equilibrium.

**2. Uncertainty assessment**
I am not convinced that all potential sources of uncertainty are represented in the significantly reduced overall uncertainty presented in Figure 6. In particular, I'm thinking about the 17 % uncertainty in the ClOOCl reference cross sections that you mention on page 8, line 14. When I translate the uncertainty in the $B$ param-eter that accounts for this (given on page 8, line 17), I arrive at approximately $\pm$ 40 % uncertainty in $K_{eq}$, which is considerably larger than the uncertainty range given in Figure 6. There, it looks as if the blue uncertainty ranges only repre-sent the statistical uncertainties in your experiment (i.e. the scatter between the individual white circles in that figure represented in the $\pm$ 25 uncertainty in $B$

given in Equation 3), and the statement about the trend to not explicitly include uncertainties from reference cross sections on page 6, line 21, seems to support that interpretation. In the work by Hume et al. (2015), there is a clear statement that the systematic uncertainties in cross sections to convert $K_{abs}$ to $K_{eq}$ are much more important than their experimental standard deviations, and their error propagation calculations arrive at an upper uncertainty limit from this is 36 % on the experimental $K_{eq}$, which appears to be a very realistic number. Note that the 17 % uncertainty in the ClOOCl cross section alone translates directly into a 17 % uncertainty in the ClOOCl concentration fitted to a particular spectrum, and in turn into a systematic 17 % uncertainty in $K_{eq}$ via your Equation 1. The two studies by Papanastasiou et al. (2009, currently the JPL recommended value and used in both, your study and Hume et al.) and Lien et al. (2009; 17 % higher) used different methods to measure absolute cross sections: Papanastasiou et al. infer the absolute value from the reaction stoichiometry and the experimental absorbance at isosbestic wavelength, while Lien et al. use a known quantity of light to attenuate a molecular beam of ClOOCl. To date, no convincing evidence has been presented that proves either one of the results right or wrong. Therefore, the 17 % is not a statistical one or two sigma uncertainty, but a highly systematic one. As long as this is not resolved, it is therefore impossible to reduce the uncertainty in $K_{eq}$ below this value with any method relying on the ClOOCl cross sections. And because it is systematic, it really has to be added to the blue ranges representing the statistical uncertainties in your Figure 6, because if the peak ClOOCl absorption cross section was 17 % higher than the value you actually used in your calculations, all your points and the ones from Hume et al. would simultaneously down by 17 %. Of course, if the uncertainties in the ClOOCl cross sections were reduced in future studies, this would immediately also reduce the uncertainty in $K_{eq}$, which I think would be a fair and valuable statement to make.

It is true that the ClOOCl cross section directly impacts $K_{eq}$, but to be clear, Hume

et al. actually did not incorporate the ClOOCl cross section uncertainty into their reported $K_{eq}$. Hume et al. used the larger uncertainty in the $B$ parameter derived when considering the ClOOCl cross section uncertainty only when they assigned uncertainties to their enthalpy calculations. For consistency, we followed the same approach in our manuscript. Otherwise, we would be defining the uncertainty in $K_{eq}$ in a manner that is inconsistent not only with the recent Hume paper but all other published laboratory ClOOCl $K_{eq}$ papers that covered a range of temperatures, none of which include the uncertainty from the reference cross sections. We state on page 6 line 19 of the revised manuscript (page 6 line 21 of the original manuscript) that we are taking this approach, and on page 8 line 23 of the revised manuscript (page 8 line 17 of the original manuscript), we provide the larger uncertainty range in the $B$ parameter found when including the ClOOCl cross section uncertainty. We have also now added additional detail to Table 1 such that if the ClOOCl or ClO cross sections are revised in the future, our $K_{eq}$ values can be recalculated.
* * *

---

## Author Comment (AC3) · 31 Jan 2019

We would like to thank the referee for his detailed and constructive comments and have revised the manuscript accordingly. The reviewer's comments are presented below in **bold** text and our responses to the reviewer appear in plain text.

**1. Please show an enlargement of residual spectra (e.g., as in Figure 2) with examples calculated at the 1-sigma ranges of the uncertainties. This is especially important for the results at lowest temperatures where ClO absorbances are**

[Figure]

**smallest relative to total absorbance due to the [ClO]-squared nature of $K_{eq}$.**

The figure requested is shown below (Figure 1) for two independent experiments in which the calculated $K_{eq}$ approximates the 1-sigma uncertainties. Note that the Y axes are different between the two panels. Again, as in Figure 2 of the manuscript, OClO and $Cl_2O_3$ are included in the fit, but excluded from the visualization as their formation is suppressed by the operation of the reaction cell at 200 K and would appear as flat lines at the 0 A.U. line of each panel.

**2. I think it would be very useful to plot the results of the $K_{eq}$ calculations versus inverse temperature of individual replicates prior to temperature averaging. This will help illustrate the reproducibility of the runs (i.e., precision) and give a better sense for the density of replicates at specific temperatures. Note, also, that it is more appropriate to average log($K_{eq}$) values from individual replicates than to average $K_{eq}$ values (as described on Page 5, line 25) to avoid a systematic bias of a few percent due to the exponential nature of $K_{eq}$.**

In this case the difference is minimal, but we agree that it is more appropriate to fit the $K_{eq}$ values from individual replicates rather than the binned data. A third law fit of all independent measurements produces a $B$ parameter value of 8528 K, which represents a change of 0.059% from the value of 8533 K reported in the original manuscript. We have revised the manuscript throughout to reflect this value.

A new Figure 3 now appears in the manuscript and is replicated below as Figure 2 of this response. In this figure, the independent experimental results are indicated as small orange circles and are used to determine the black fit line. Note that the density of orange points is obscured for some temperatures where the scatter is small, and thus the updated Table 1 should be referenced.

**3. Because of the strong dependence of $K_{eq}$ on temperature, a more detailed description of temperature variations and accuracy is essential. Please show (or**

**describe) how temperature varies axially and radially within the measurement cell during a given replicate. Also, it would be useful to know how much temperature varies with time over the course of a particular replicate. I am a little concerned that a single-point measurement of temperature in the center of a measurement cell may not be adequate for a quantitative assessment of uncertainties (e.g., note that a 1.0 degree variation in temperature translates into a 15% variance in $K_{eq}$). Presumably the uniformity of temperature has been carefully measured and documented at various temperatures. If so, presentation of such evidence will greatly strengthen the case that this new measurement can be used to reduce uncertainties in the JPL assessment.**

As shown in Figure 1 of the original manuscript, the temperature measurements were conducted at several key points (including immediately prior to the entrance of the gas mixture to the absorption cell and at the halfway point of the absorption cell). For all results reported in this work, the temperature difference between these points was less than 1 K. Radial measurements of the temperature gradient were also performed. Temperature differences of less than 0.5 K were observed between the wall region and the center of flow. We note that, to optimize lamp signal (i.e., to avoid clipping the light), the thermistor junction in the absorption cell was maintained at an intermediate position between the wall and the center of flow for all experimental results reported in this work.

Absorbance and equilibrium cell temperatures can be maintained at a near-constant temperature for an indefinite period of time and are well-insulated from any interference from the surrounding environment. The insulating material used here, cryogel-Z, is an extremely high-quality insulating material. There was never any condensation of water on the experiment, even when operating at 203 K (the lowest attainable temperature of the experiment) for several hours. Gas temperature was observed to remain static (variation within the noise levels of the thermistor ADC) over the course of a single sample acquisition (3 minutes time).

**4. I appreciate the rationale for co-varying pressure and temperature; however, given the possibility of systematic biases due to pressure (e.g. secondary reactions), it would be useful to know if any detailed measurements with varying pressure were carried out for a fixed temperature. On Page 6, starting on Line 10, the authors state "The precision of repeated measurements conducted at the same temperature but varied flow rates and pressures did not statistically deviate from the precision from temperature dependence alone." Over what range (or percentage) were flow rates and pressures varied for a fixed temperature? Or does this refer to unintended variations that may have occurred over the course of a particular replicate?**

Experiments were performed at a selection of pressures in order to verify asymptotic equilibrium behavior at a fixed temperature. It was typical to scan pressures by $\pm 20\%$ of the target pressure when evaluating conditions prescribed by the kinetic model. Pressure and flow rates were maintained at a constant value during sample acquisition.

**5. The authors need to show a more detailed error analysis that traces the various sources of error (e.g., from spectral fitting, temperature, and errors in rate parameters for interfering secondary reactions, if relevant). They should also include an assessment of potential systematic errors (such as those described above). They could expand Table 1 to include these errors. I am not sure that the standard deviation values listed in Table 1 are uniformly illustrative - for example, there is no way that a 0.3% standard deviation from two independent measurements at 285.1 K is representative of the true precision when the standard deviation is 10% for the 8 replicates at 253.3 K.**

Page 6 of the original manuscript contains a detailed assessment of error from spectral fitting, temperature-dependence, and other obvious potential systematic errors. As for secondary reactions such as the formation of OClO and higher oxides of chlorine, these are highly suppressed in our system due to the operation of the reaction cell at cold temperatures. The Figure 2 caption states that OClO and $Cl_2O_3$ concentrations were

small (and even if present, they would not compromise the experiment, because they are included in the spectral fits). We also point out that we essentially quench the Cl + ClOOCl reaction pathway by operating the experiment with an excess of ozone.

We have revised the method by which we calculate $K_{eq}$, now evaluating each independent measurement separately in an ordinary least squares fit. Because of this, metrics required for the reproduction of the weighted-least squares fit of the binned data, such as standard deviations of similar-temperature replicates, have been removed from Table 1.

**Minor comments**
**6. Abstract/conclusions. The authors should report the value of $K_{eq}$ over the temperature range 230 to 299 K, reflecting the range over which they have calculated their experimental averages. Alternatively, if they want to claim significance for a measurement at 288 K then they should report a value that is measured over the range 285.5 to 290.5 K (assuming a similar 5 degree average).**

We now calculate $K_{eq}$ using every independent replicate, spanning the temperature range of 228 –301 K.

**7. Page 2, line 28. Please elaborate on ". . .optimization of target chemistry." What, specifically, was optimized?**

The various flow sections shown in Figure 1 are operated at the optimal conditions for achieving thermal equilibrium. This sentence is simply an introduction to the flow section descriptions that follow.

**8. Page 2, lines 30-31. Discuss whether or not you expect discharge of oxygen to produce $O_2(^1\Delta)$, and if so, how you might expect reactions of this specie to impact your results.**

$N_2/O_2/O_3$ addition is performed downstream of the microwave discharge, which maintained a constant salmon color when chlorine was not injected and a constant deep

purple color when $Cl_2$ was injected. The color of the discharge did not change when $O_2/O_3$ flows were turned on. When pressure was scanned above 533 mbar, the discharge was observed to turn white, indicating backflow of nitrogen from the injector port. All reported experiments were conducted below 333 mbar.

The exact placement of the microwave cavity relative to the $O_3$ addition port varied for some experiments, as did the size of the discharge depending on flow conditions, but during operating conditions as reported in this work (100 − 333 mbar), we observed no interfering absorbers in the UV spectra (e.g., residual traces were homoskedastic). We observed no evidence of interference from excited oxygen or nitrogen species produced in the microwave discharge.

**9. Page 3, line 30. Please list your carrier gas flow rates and residence times in each of the cells.**

Flow rates for the carrier gases ranged between ~1.0 −1.8 L/min and residence times in the absorption cell ranged between ~1 −11 seconds, depending on pressure and temperature. These values have been added to the manuscript on page 3 line 30 and page 4 line 2.

**10. Page 7, lines 31–33. It might be helpful to include a representative 1 sigma uncertainty bar on the results from the February 3, 2000, SOLVE/THESEO ER-2 flight in Figure 5. Please note whether "measurement uncertainties" for those data points refer to uncertainties (or variability) in measured concentrations of ClO and ClOOCl, uncertainties (or variability) of measured temperature, or both. This could also be illustrated with the use of vertical (for concentrations) and horizontal (for temperature) error bars.**

Including the $2\sigma$ uncertainties, all of the SOLVE/THESEO data points overlap the $2\sigma$ uncertainty range of our fit. This fact is now stated in the manuscript on page 8 lines 3 –5.

**11. Page 9, line 8. You might elaborate on how, specifically, the uncertainties in JPL recommended $K_{eq}$ can be reduced. Should results of previous experiments be discounted by the JPL panel? Or should results of various experiments over the years be averaged and weighted according to errors reported at the time?**

The current uncertainty envelope is derived from the minimum uncertainty required to envelop the independent results of Cox and Hayman (1988), Nickolaisen et al. (1994), and Hume et al. (2015). Our results exhibit significantly less scatter than the two earlier studies. Though we do not presume to tell the JPL panel how to evaluate the uncertainty of the ClO/ClOOCl equilibrium constant, a similar approach to the one conducted to determine prescribed uncertainty for the 2015 data evaluation using our work instead of the older studies would produce a significantly smaller uncertainty.

[Figure]

[Figure]

**Fig. 1.** deconvolution of raw absorbance spectra into individual gas components.

Fig. 2. All independent experimental values of the thermal equilibrium as a function of 1000/T

---

## Author Response (AR1)

We would like to thank the referees for their detailed and constructive comments. We have revised the manuscript accordingly and our responses to the reviewers are below.

**Referee #1:  J.R. Barker**

**1. A quibble with the third-law analysis carried out in this paper is that it assumes that Keq is a pure exponential function, Keq = A exp(B/T), while theoretical calculations predict that Keq deviates slightly from a pure exponential. The approach taken in this paper differs from the third-law analysis carried out by the NASA/JPL Panel for Data Evaluation. The Panel evaluates every individual data point to determine the reaction enthalpy change at 0 K, ΔHr(0), extracted from the measured Keq(T) at each temperature and then determines the average. ΔHr(298) is then computed from the calculated reaction entropies and average ΔHr(0). From ΔHr(298), one can then obtain the standard enthalpy of formation, ΔHf(298), for ClOOCl. Although more laborious, the approach followed by the panel does not assume a pure exponential function and is thought to be more accurate. From the Panel's approach, the combined data set consisting of the new data from the present paper and the older data from Hume et al. (2015) gives ΔHr(298) = 130.1 kJ/mol, which is 0.2 kJ/mol higher than the result reported in the manuscript, based on the pure exponential function.**

We appreciate the approach described by the reviewer and the accuracy of this method for use in the NASA/JPL Data Panel Evaluation. However, to maintain consistency between our results and the method employed in prior papers (e.g., Hume et al. 2015), we have chosen to use the enthalpy values at 298 K for the calculations. Note that we now determine B from our unbinned data, and the calculated ΔHf(298) for ClOOCl is 130.0 kJ/mol, now in even better agreement with JPL-15 recommendation of 130.1 kJ/mol.

To facilitate alternate derivations of reaction enthalpies, all independent measurements are now reported in Table 1.

**2. In the abstract, it would be useful to state that the experimental errors are ±1 σ.**

The abstract now reads:
A third law fit of the equilibrium values determined from the experimental data provides the expression: Keq = 2.16×10−27e (8528 ± 25 K/T) cm3 molecule−1 (1σ  uncertainty).

**3. In addition to the binned values for Keq(T), which are reported in Table 1, all of the individual Keq measurements should be reported in Supporting Information.**

All independent measurements are now reported in Table 1.

**4. In Figures 4 and 6, the error bounds are functions of temperature. It would be useful to report the functional forms and parameters for the error bounds in Supporting Information.**

The error bounds of Figure 4 in the original manuscript (now Figure 5) are of the form: A exp (B±ε / T) from equation 2, where ε is the appropriate uncertainty interval. We now explicitly mention the ε value on page 7 line 9 of the revised manuscript.

**Referee #2:  M. von Hobe**

**1. Add more experimental information**
**I have little doubt that the authors have carefully checked the conditions of each experiment to ensure that thermal equilibrium between ClO and ClOOCl is actually established once the gas mixture passes into the absorption cell. Nevertheless, for the sake of transparency, it would be good to provide some actual numbers for flow rates and residence times in the different parts of the apparatus. Ideally, a complete set of experimental conditions (initial concentrations, pressure, $T_{ReactionCell}$, $T_{ColdT rap}$, $T_{EquilibriumCell}$) and selected results (maybe even some raw spectra) could be provided as an electronic supplement that goes beyond the summary given in Table 1.**

Table 1 has now been revised to include all of the experimental runs and not simply the binned data, and details on carrier gas flows and residence time have been added in section 2. We note that a critical element of the experimental design and operation was ensuring that ClO and ClOOCl were in equilibrium in the Absorption Cell when the measurements were made. As discussed in the manuscript, the kinetic model defined the experimental conditions (e.g., temperatures, pressures, concentrations) in which equilibrium would be achieved in the laboratory setup, and we operated experimentally across a range of conditions about the optimal starting conditions predicted by the model. In particular, we note that the same Keq values were obtained when increasing or decreasing residence times in the equilibrium and absorption cells, providing confidence that ClO and ClOOCl were indeed in equilibrium.

**2. Uncertainty assessment**
**I am not convinced that all potential sources of uncertainty are represented in the significantly reduced overall uncertainty presented in Figure 6. In particular, I'm thinking about the 17 % uncertainty in the ClOOCl reference cross sections that you mention on page 8, line 14. When I translate the uncertainty in the B parameter that accounts for this (given on page 8, line 17), I arrive at approximately +/- 40 % uncertainty in Keq, which is considerably larger than the uncertainty range given in Figure 6. There, it looks as if the blue uncertainty ranges only represent the statistical uncertainties in your experiment (i.e. the scatter between the individual white circles in that figure represented in the +/- 25 uncertainty in B given in Equation 3), and the statement about the trend to not explicitly include uncertainties from reference cross sections on page 6, line 21, seems to support that interpretation. In the work by Hume et al. (2015), there is a clear statement that the systemati uncertainties in cross sections to convert Kabs to Keq are much more important than their experimental standard deviations, and their error propagation calculations arrive at an upper uncertainty limit from this is 36 % on the experimental Keq, which appears to be a very realistic number. Note that the 17 % uncertainty in the ClOOCl cross section alone translates directly into a 17 % uncertainty in the ClOOCl concentration fitted to a particular spectrum, and in turn into a systematic 17 % uncertainty in Keq via your Equation 1. The two studies by Papanastasiou et al. (2009, currently the JPL recommended value and used in both, your study and Hume et al.) and Lien et al. (2009; 17 % higher) used different methods to measure absolute cross sections: Papanastasiou et al. infer the absolute value from the reaction stoichiometry and the experimental absorbance at isosbestic wavelength, while Lien et al. use a**

**known quantity of light to attenuate a molecular beam of ClOOCl. To date, no convincing evidence has been presented that proves either one of the results right or wrong. Therefore, the 17 % is not a statistical one or two sigma uncertainty, but a highly systematic one. As long as this is not resolved, it is therefore impossible to reduce the uncertainty in Keq below this value with any method relying on the ClOOCl cross sections. And because it is systematic, it really has to be added to the blue ranges representing the statistical uncertainties in your Figure 6, because if the peak ClOOCl absorption cross section was 17 % higher than the value you actually used in your calculations, all your points and the ones from Hume et al. would simultaneously down by 17 %. Of course, if the uncertainties in the ClOOCl cross sections were reduced in future studies, this would immediately also reduce the uncertainty in Keq, which I think would be a fair and valuable statement to make.**

It is true that the ClOOCl cross section directly impacts Keq, but to be clear, Hume et al. actually did not incorporate the ClOOCl cross section uncertainty into their reported Keq. Hume et al. used the larger uncertainty in the B parameter derived when considering the ClOOCl cross section uncertainty only when they assigned uncertainties to their enthalpy calculations. For consistency, we followed the same approach in our manuscript. Otherwise, we would be defining the uncertainty in Keq in a manner that is inconsistent not only with the recent Hume paper but all other published laboratory ClOOCl Keq papers that covered a range of temperatures, none of which include the uncertainty from the reference cross sections. We state on page 6 line 19 of the revised manuscript (page 6 line 21 of the original manuscript) that we are taking this approach, and on page 8 line 23 of the revised manuscript (page 8 line 17 of the original manuscript), we provide the larger uncertainty range in the B$parameter found when including the ClOOCl cross section uncertainty. We have also now added additional detail to Table 1 such that if the ClOOCl or ClO cross sections are revised in the future, our Keq values can be recalculated.

**Referee #3:  D. Toohey**
**1. Please show an enlargement of residual spectra (e.g., as in Figure 2) with examples calculated at the 1-sigma ranges of the uncertainties. This is especially important for the results at lowest temperatures where ClO absorbances are smallest relative to total absorbance due to the [ClO]-squared nature of Keq.**

The figure requested is shown below for two independent experiments in which the calculated $K_{eq}$ approximates the 1-sigma uncertainties. Note that the Y axes are different between the two panels. Again, as in Figure 2 of the manuscript, OClO and Cl2O3 are included in the fit, but excluded from the visualization as their formation is suppressed by the operation of the reaction cell at 200 K and would appear as flat lines at the 0 A.U. line of each panel.

[Figure]

**The figure requested is shown below for two independent experiments in which the calculated K$_{eq}$ approximates the 1-sigma uncertainties. Note that the Y axes are different between the two panels. Again, as in Figure 2 of the manuscript, OClO and Cl2O3 are included in the fit, but excluded from the visualization as their formation is suppressed by the operation of the reaction cell at 200 K and would appear as flat lines at the 0 A.U. line of each panel.**

In this case the difference is minimal, but we agree that it is more appropriate to fit the Keq values from individual replicates rather than the binned data. A third law fit of all independent measurements produces a B parameter value of 8528 K, which represents a change of 0.059% from the value of 8533 K reported in the original manuscript. We have revised the manuscript throughout to reflect this value.

A new Figure 3 now appears in the manuscript and is replicated below. In this figure, the independent experimental results are indicated as small orange circles and are used to determine the black fit line. Note that the density of orange points is obscured for some temperatures where the scatter is small, and thus the updated Table 1 should be referenced.

[Figure]

**3. Because of the strong dependence of Keq on temperature, a more detailed description of temperature variations and accuracy is essential. Please show (or describe) how temperature varies axially and radially within the measurement cell during a given replicate. Also, it would be useful to know how much temperature varies with time over the course of a particular replicate. I am a little concerned that a single-point measurement of temperature in the center of a measurement cell may not be adequate for a quantitative assessment of uncertainties (e.g., note that a 1.0 degree variation in temperature translates into a 15% variance in Keq). Presumably the uniformity of temperature has been carefully measured and documented at various temperatures. If so, presentation of such evidence will greatly strengthen the case that this new measurement can be used to reduce uncertainties in the JPL assessment.**

As shown in Figure 1 of the original manuscript, the temperature measurements were conducted at several key points (including immediately prior to the entrance of the gas mixture to the absorption cell and at the halfway point of the absorption cell). For all results reported in this work, the temperature difference between these points was less than 1 K. Radial measurements of the temperature gradient were also performed. Temperature differences of less than 0.5 K were observed between the wall region and the center of flow. We note that, to optimize lamp signal (i.e., to avoid clipping the light), the thermistor junction in the absorption cell was maintained at an intermediate position between the wall and the center of flow for all experimental results reported in this work.

Absorbance and equilibrium cell temperatures can be maintained at a near-constant temperature for an indefinite period of time and are well-insulated from any interference from the surrounding environment.  The insulating material used here, cryogel-Z, is an extremely high-quality insulating material. There was never any condensation of water on the experiment, even when operating at 203 K (the lowest attainable temperature of the experiment) for several hours.  Gas temperature was observed to remain static (variation within the noise levels of the thermistor ADC) over the course of a single sample acquisition (3 minutes time).

**4. I appreciate the rationale for co-varying pressure and temperature; however, given the possibility of systematic biases due to pressure (e.g. secondary reactions), it would be useful to know if any detailed measurements with varying pressure were carried out for a fixed temperature. On Page 6, starting on Line 10, the authors state "The precision of repeated measurements conducted at the same temperature but varied flow rates and pressures did not statistically deviate from the precision from temperature dependence alone." Over what range (or percentage) were flow rates and pressures varied for a fixed temperature? Or does this refer to unintended variations that may have occurred over the course of a particular replicate?**

Experiments were performed at a selection of pressures in order to verify asymptotic equilibrium behavior at a fixed temperature. It was typical to scan pressures by +/- 20% of the target pressure when evaluating conditions prescribed by the kinetic model. Pressure and flow rates were maintained at a constant value during sample acquisition.

**5. The authors need to show a more detailed error analysis that traces the various sources of error (e.g., from spectral fitting, temperature, and errors in rate parameters for interfering secondary reactions, if relevant). They should also include an assessment of potential systematic errors (such as those described above). They could expand Table 1 to include these errors. I am not sure that the standard deviation values listed in Table 1 are uniformly illustrative - for example, there is no way that a 0.3% standard deviation from two independent measurements at 285.1 K is representative of the true precision when the standard deviation is 10% for the 8 replicates at 253.3 K.**

Page 6 of the original manuscript contains a detailed assessment of error from spectral fitting, temperature-dependence, and other obvious potential systematic errors. As for secondary reactions such as the formation of OClO and higher oxides of chlorine, these are highly suppressed in our system due to the operation of the reaction cell at cold temperatures. The Figure 2 caption states that OClO and Cl2O3 concentrations were small (and even if present, they would not compromise the experiment, because they are included in the spectral fits). We also point out that we essentially quench the Cl + ClOOCl reaction pathway by operating the experiment with an excess of ozone.

We have revised the method by which we calculate $K_{eq}$, now evaluating each independent measurement separately in an ordinary least squares fit. Because of this, metrics required for the reproduction of the weighted-least squares fit of the binned data, such as standard deviations of similar-temperature replicates, have been removed from Table 1.

**Minor comments**
**6. Abstract/conclusions. The authors should report the value of Keq over the temperature range 230 to 299 K, reflecting the range over which they have calculated their experimental averages. Alternatively, if they want to claim significance for a measurement at 288 K then they should report a value that is measured over the range 285.5 to 290.5 K (assuming a similar 5 degree average).**

We now calculate Keq using every independent replicate, spanning the temperature range of 228 – 301 K.

**7. Page 2, line 28. Please elaborate on ". . .optimization of target chemistry." What, specifically, was optimized?**

The various flow sections shown in Figure 1 are operated at the optimal conditions for achieving thermal equilibrium. This sentence is simply an introduction to the flow section descriptions that follow.

**8. Page 2, lines 30-31. Discuss whether or not you expect discharge of oxygen to produce O2(singlet delta), and if so, how you might expect reactions of this specie to impact your results.**

N2/O2/O3 addition is performed downstream of the microwave discharge, which maintained a constant salmon color when chlorine was not injected and a constant deep purple color when chlorine was injected. The color of the discharge did not change when O2/O3 flows were turned on. When pressure was scanned above 533 mbar, the discharge was observed to turn white, indicating backflow of nitrogen from the injector port. All reported experiments were conducted below 333 mbar.

The exact placement of the microwave cavity relative to the O3 addition port varied for some experiments, as did the size of the discharge depending on flow conditions, but during operating conditions as reported in this work (100 – 333 mbar), we observed no interfering absorbers in the UV spectra (e.g., residual traces were homoskedastic). We observed no evidence of interference from excited oxygen or nitrogen species produced in the microwave discharge.

**9. Page 3, line 30. Please list your carrier gas flow rates and residence times in each of the cells.**

Flow rates for the carrier gases ranged between ~1.0 – 1.8 L/min and residence times in the absorption cell ranged between ~1 – 11 seconds, depending on pressure and temperature. These values have been added to the manuscript on page 3 line 30 and page 4 line 2.

**10. Page 7, lines 31-33. It might be helpful to include a representative 1 sigma uncertainty bar on the results from the February 3, 2000, SOLVE/THESEO ER-2 flight in Figure 5. Please note whether "measurement uncertainties" for those data points refer to uncertainties (or variability) in measured concentrations of ClO and ClOOCl, uncertainties (or variability) of measured temperature, or both. This could also be illustrated with the use of vertical (for concentrations) and horizontal (for temperature) error bars.**

Including the 2-sigma uncertainties, all of the SOLVE/THESEO data points overlap the 2-sigma uncertainty range of our fit. This fact is now stated in the manuscript on page 8 lines 3 – 5.

**11. Page 9, line 8. You might elaborate on how, specifically, the uncertainties in JPL recommended Keq can be reduced. Should results of previous experiments be discounted by the JPL panel? Or should results of various experiments over the years be averaged and weighted according to errors reported at the time?**

The current uncertainty envelope is derived from the minimum uncertainty required to envelop the independent results of Cox and Hayman (1988), Nickolaisen et al. (1994), and Hume et al. (2015). Our results exhibit significantly less scatter than the two earlier studies. Though we do not presume to tell the JPL panel how to evaluate the uncertainty of the ClO/ClOOCl equilibrium constant, a similar approach to the one conducted to determine prescribed uncertainty for the 2015 data evaluation using our work instead of the older studies would produce a significantly smaller uncertainty.

**Referee #4:  Anonymous**

**The microwave discharge that generates atomic chlorine will also be a strong source of VUV radiation due to the strong Cl resonance lines. Since the ozone is introduced only 2.5 cm downstream, one would expect significant photolysis of the ozone, generating C1 O(1D), O(3P), O2(a1delta), etc. Also the discharge that generates ozone will also form copious amounts of the singlet states of O2. Does the chemical modeling of this system show that none of these reactive species are causing trouble downstream?**

There was no evidence that secondary reactions interfered with our measurements. O3 was transported ~3 m subsequent to generation prior to injection into the experiment. O2(singlet delta) would certainly be quenched prior to injection over this length scale. We observed no evidence of interfering species in the absorbance spectra after fitting for known species (e.g., residual was homoskedastic).

During initial setup and calibration of the apparatus, experiments were performed in which the microwave cavity was placed at varying distances from the ozone injection port and ClO absorbances were quantified. The selected position of the cavity, ~2.5 cm upstream of the ozone injector, was found to be the optimal position for production of ClO (competing with the recombination of Cl). Please our response to Referee #3 for additional details on this topic.

**Typical gas flow rates or the velocities of the flows at the different pressures should be given**

We have revised page 3 line 30 and page 4 line 2 of the manuscript to provide this information.

**Page 4, line 5: What is the resolution (FWHM) for this slit width?**

The grating and slit width combination produce a resolution of ~0.3 nm. Page 4, line 5 of the revised manuscript now includes this information.

**Page 4, line 10: What are "dark spectra" and how are they used in subsequent spectra?**

Prior to any day of experiments, a spectrum is collected in which the UV lamp is off. This spectrum is subsequently subtracted from all subsequent spectra to provide a correction for dark current in the photodetector.

**The two runs shown in Fig. 2 can be used to calculate two equilibrium constants: for 230K, K = 1.84E(-11) which is 34% lower than Equation 3 predicts; for 300K, K = 6.52E(-15) which is 34% higher than Eq.3. Are these typical deviations?**

The spectra presented in Figure 2 were selected because they both featured very similar concentrations of O3, allowing for comparison of the fit residual on the same scale. While these spectra are both members of the experimental ensemble reported, they are far from the best examples we could have chosen here to produce a Keq in close agreement with the mean reported value. A full list of independent experiments is now presented in Table 1.

**The treatment of the experimental data by "binning" is not the best use of the experimental data. If I understand this manuscript correctly, all values of K measured within a 5 K range of temperatures are averaged (average of K, or ln(K)?) and then listed in Table 1 along with the averaged temperature (average T or 1/T?). For example, if measurements were done at 250 and 255 K, one should get, according to Equation 3, K(250) = 1.438E(-12) and K(255) = 7.364E(-13). The average of ln(K) of these two K's gives -27.602 or K(ave) = 1.029E(-12), which is close to the value predicted by Eq. 3 for T = 252.5 K. But then using the deviations of these two values of K from the average K to get an estimate of the standard deviation of the measurements is not valid. There is no information about the random measurement errors in these two numbers. They are different because they were measured at two different temperatures. Supposed one has only two measurements in one bin that were are different temperatures but, due to random errors, gave almost the same value for K. The calculated "standard deviation" would now be very small and the weighting factor very large. That is not right. The numbers in the last column in Table 1 should be eliminated. It would be better to do the least squares fitting using all 114 experimental measurements with equal weighting. Then the deviations of the experimental K values from the least squares fit would give information about the precision of the measurements.**

In this case the difference is minimal, but we agree that it is more appropriate to fit the Keq values from individual replicates rather than the binned data. A third-law fit of all independent measurements produces a B parameter value of 8528 K, which represents a change of 0.059% from the value of 8533 K reported in the original manuscript. We have revised the manuscript throughout to reflect this value.

**A plot of these 114 deviations vs. 1/T would be useful. As mentioned, this will probably show increasing deviations at both the upper and lower limits of temperature.**

The reviewer is correct that there are increasing deviations at the upper and lower limits of temperature. A new Figure 3 of all independent results is now included in the manuscript.

**Certainly the experimental values of the 114 measurements should be preserved, either in a table in the manuscript or as supplemental material.**

These independent measurements are now included in Table 1.

[revised manuscript text omitted]

---

## Author Response (AR2)

We have revised the manuscript according to the editor's suggestions and our responses are below.

**\* The use of "binned" data in the figures following Fig. 3 requires some clarification in the text. I suggest that the authors consider including revised figures including "unbinned" data in either the main manuscript or as supplementary material.**

Data are no longer presented in 5 K bins. We added a new Figure 3b, which shows the ratio of all of our experimental Keq values to the third law fit. The error shading in this figure now allows for a straightforward comparison with the later figures. For clarity in figure presentation for Figures 4, 5, and 7, we now present our data in 3 K averages, which is the smallest averaging range in which each data point is averaged with at least one other data point. Without averaging our data, the data from the other studies are difficult to see on these later plots. We emphasize that all quantitative analyses in this study were conducted using the full data set of Keq measurements as shown in Figure 3 and Table 1.

**\* The residual UV spectra included in Figure 2 should be plotted on an expanded scale to more clearly show the quality of the subtractions and potential systematic deviations, if any.**

The residual UV spectra presented in Figure 2 have now been magnified to more clearly show the quality of the spectral fits. In addition to this change, we have revised the spectral fitting method, motivated by the desire to better quantify OClO concentrations for ruling out secondary chemistry, and this is reflected in the text and the new version of Figure 2. Specifically, to better constrain OClO, we now fit this molecule at longer wavelengths (310– 350 nm), where the cross sections are much greater and have smaller uncertainties. ClO is now fit using the vibrational structure only, which also provides improved results. ClOOCl is fit as before, constraining ClO to the value obtained in the differential absorption method. The $B$ parameter from the third law fit changed slightly from 8528 K to 8527 K. The second law fit parameters also changed; however, not beyond the 1σ uncertainty reported previously. As demonstrated in Figure 2, OClO is small for all runs in this study.

**\* Table 1: The table should be expanded to include any relevant physical conditions for each measurement but more importantly the concentrations of ClO, Cl2O2, and OClO(if observed) and their uncertainties for each experiment. The precision of the determined Keq should also be provided.**

Table 1 now includes the concentrations of ClO, ClOOCl, and OClO (when it is observed) and uncertainties in these values, as derived from an additional analysis, which is described in the text of the revised manuscript.

**\* Table 1: The temperatures quoted in table 1 to 0.01 C is unrealistic considering the accuracy of the temperature measurement in the experiment. Appropriate temperature values should be quoted and possibly used in the data analysis.**

In our original analysis, we retained and propagated an additional decimal place beyond the last significant figure in the analog-to-digital conversion of the temperature reading, choosing to round the final result after all Keq calculations were completed rather than round in the intermediate steps. Accordingly, to exactly reproduce our second law fit results, the temperature had to be reported to two decimals. We now truncate our temperatures to 0.1 C and propagate these values into all calculations of Keq. We find that the third law *B* parameter value is identical between both treatments; however, the second law parameters have adjusted slightly, though not significantly.

**\* Conclusions: The text for comparing the JPL recommended uncertainty and the statistical uncertainty obtained in the present work needs to be revised to clearly state that the JPL value was not derived from a statistical analysis.**

We now state unambiguously that the JPL recommended uncertainty is not derived from a statistical analysis of prior works.